# PAICS/DYRK3 Multienzyme Interactions as Coregulators of Purinosome Formation and Metabolism on Radioresistance in Oral Squamous Cell Carcinoma

**DOI:** 10.3390/ijms242417346

**Published:** 2023-12-11

**Authors:** Chin-Sheng Huang, Ming-Shou Hsieh, Vijesh Kumar Yadav, Yang-Che Wu, Shao-Cheng Liu, Chi-Tai Yeh, Mao-Suan Huang

**Affiliations:** 1Department of Dentistry, Taipei Medical University-Shuang Ho Hospital, New Taipei City 235, Taiwan; 09520@s.tmu.edu.tw (C.-S.H.); dr7d9eddd@gmail.com (M.-S.H.); vijeshp2@gmail.com (V.K.Y.); yangchewu@tmu.edu.tw (Y.-C.W.); 2School of Dentistry, College of Oral Medicine, Taipei Medical University, Taipei City 110, Taiwan; 3Department of Dentistry and Oral Health, Taipei Medical University-Shuang Ho Hospital, New Taipei City 235, Taiwan; 4Department of Otolaryngology-Head and Neck Surgery, Tri-Service General Hospital, National Defense Medical Center, Taipei City 114, Taiwan; m871435@mail.ndmctsgh.edu.tw; 5Department of Medical Research & Education, Taipei Medical University-Shuang Ho Hospital, New Taipei City 235, Taiwan; 6Continuing Education Program of Food Biotechnology Applications, College of Science and Engineering, National Taitung University, Taitung 950, Taiwan

**Keywords:** oral squamous cell carcinoma, DYRK3, PAICS, radioresistance, purinosome

## Abstract

Oral squamous cell carcinoma (OSCC) is a prevalent type of oral cancer. While therapeutic innovations have made strides, radioresistance persists as a significant hindrance in OSCC treatment. Despite identifying numerous targets that could potentially suppress the oncogenic attributes of OSCC, the exploration of oncogenic protein kinases for cancer therapy remains limited. Consequently, the functions of many kinase proteins in OSCC continue to be largely undetermined. In this research, we aim to disclose protein kinases that target OSCC and elaborate their roles and molecular mechanisms. Through the examination of the kinome library of radiotherapy-resistant/sensitive OSCC cell lines (HN12 and SAS), we identified a key gene, the tyrosine phosphorylation-regulated kinase 3 (DYRK3), a member of the DYRK family. We developed an in vitro cell model, composed of radiation-resistant OSCC, to scrutinize the clinical implications and contributions of DYRK3 and phosphoribosylaminoimidazole carboxylase and phosphoribosylaminoimidazolesuccinocarboxamide synthase (PAICS) signaling in OSCC. This investigation involves bioinformatics and human tissue arrays. We seek to comprehend the role of DYRK3 and PAICS signaling in the development of OSCC and its resistance to radiotherapy. Various in vitro assays are utilized to reveal the essential molecular mechanism behind radiotherapy resistance in connection with the DYRK3 and PAICS interaction. In our study, we quantified the concentrations of DYRK3 and PAICS proteins and tracked the expression levels of key pluripotency markers, particularly PPAT. Furthermore, we extended our investigation to include an analysis of Glut-1, a gene recognized for its linkage to radioresistance in oral squamous cell carcinoma (OSCC). Furthermore, we conducted an in vivo study to affirm the impact of DYRK3 and PAICS on tumor growth and radiotherapy resistance, focusing particularly on the role of DYRK3 in the radiotherapy resistance pathway. This focus leads us to identify new therapeutic agents that can combat radiotherapy resistance by inhibiting DYRK3 (GSK-626616). Our in vitro models showed that inhibiting PAICS disrupts purinosome formation and influences the survival rate of radiation-resistant OSCC cell lines. These outcomes underscore the pivotal role of the DYRK3/PAICS axis in directing OSCC radiotherapy resistance pathways and, as a result, influencing OSCC progression or therapy resistance. Our findings also reveal a significant correlation between DYRK3 expression and the PAICS enzyme in OSCC radiotherapy resistance.

## 1. Introduction

Oral squamous cell carcinoma (OSCC) is the most prevalent type of oral cancer, comprising 80 to 90 percent of malignant tumors found in the oral cavity. Globally, it ranks as the sixth most common cancer, with around 300,000 to 400,000 new cases diagnosed annually, primarily in developing or industrial countries with poor sanitation and economic conditions [1]. OSCC significantly impacts essential functions such as chewing, swallowing, language, and breathing, and can even pose life-threatening risks due to the distinctive physiological and anatomical characteristics of the oral cavity. Unfortunately, most OSCC patients seek medical attention when the disease has already progressed to an advanced stage, resulting in unfavorable treatment outcomes [2]. Despite decades of research, the overall effectiveness of the OSCC treatment has remained suboptimal, with a 5-year survival rate ranging from 50% to 60%. Therefore, improving therapeutic outcomes for oral cancer patients has become a key focus of clinical and basic research [3]. The implementation of standardized diagnosis and treatment protocols aims to enhance the effectiveness of treatment for malignant tumors. Additionally, individualized treatment based on these standardized approaches may further improve patient outcomes. However, cancer cells can develop resistance to radiation therapy, leading to treatment failure and poor prognoses. Updating the pathological staging of OSCC patients is crucial for developing up-to-date treatment plans that can enhance patient survival. Several genes, such as RAS, RAF, and BCL2, have been identified as being associated with radioresistance in OSCC, emphasizing the link between gene expression and resistance to radiation therapy [4]. The proper transcription and translation of specific genes are pivotal in regulating cell division, and any abnormalities in these processes can result in uncontrolled cell growth. A limited number of genes have been identified as critical in preventing, initiating, and advancing tumors, with dysfunctions or loss-of-function variants observed in various malignancies [5]. Since the 1980s, considerable progress has been made in understanding the molecular mechanisms of cancer, and protein kinases have emerged as potential targets for drug therapy. Protein kinases play crucial roles in regulating essential cellular signals and are involved in various biological processes, such as tumor cell growth, proliferation, differentiation, metabolism, apoptosis, and drug resistance/sensitivity [6]. The human genome currently contains approximately 518 known kinase genes. While these kinases are tightly regulated in normal cells, they may acquire transformative functions through mutations in disease conditions [7].

One critical signaling pathway involved in regulating cell proliferation in OSCC is the extracellular signal-regulated kinase (ERK) pathway, which encompasses mitogen-activated protein kinases (MAPKs), p38 MAPKs, and c-Jun N-terminal kinases/stress-activated protein kinases [8]. Furthermore, increased expression of DNA-dependent protein kinase (DNA-PK) has been observed in OSCC patients following radiotherapy [9]. Targeted therapies using selective kinase inhibitors have been developed for cancer treatment. However, the role of other important protein kinases in OSCC remains largely unknown, despite their potential significance in disease severity and prognosis. The DYRK (dual-specificity tyrosine-regulated kinase) family is one such group of protein kinases, evolutionarily conserved and found in organisms ranging from yeast to humans [10]. DYRKs are versatile factors that phosphorylate a wide range of proteins involved in diverse cellular processes, including cell proliferation, drug resistance, programmed cell death, and dysfunctions associated with tumorigenesis and progression [11]. Recent studies suggest that alterations in DYRK gene expression may contribute to tumorigenesis and/or disease progression, making DYRK kinases an intriguing area of research [12]. DYRKs in humans act as multifaceted elements, phosphorylating a wide array of proteins engaged in various cellular functions. They are linked with every hallmark of cancer, ranging from genomic instability to enhanced proliferation, resistance, programmed cell death, and the dysfunction of signaling pathways which are critical for tumor development and advancement. Reflecting the role of DYRK kinases in controlling cancer-related processes, there has been a growing body of research recently. These studies demonstrate changes in DYRK gene expression in tumor specimens and/or provide insights into DYRK-dependent pathways that play a role in the initiation and progression of tumors. The DYRK subfamily consists of five members: class I DYRKs (DYRK1A and DYRK1B) and class II DYRKs (DYRK2, DYRK3, and DYRK4) [13]. Exposure to radiation prompts an increase in the expression of DYRK3 (dual-specificity tyrosine phosphorylation-regulated kinase 3), which significantly impacts mitochondrial dynamics. This alteration leads to a metabolic shift and enhances tumor aggressiveness. The mechanism behind this involves both the mammalian target of rapamycin (mTOR) and DYRK3. These proteins have the ability to activate transcription factor 4 (ATF4), which in turn mediates the transcription of methylenetetrahydrofolate dehydrogenase 2 (MTHFD2). MTHFD2 is critical for producing 10-fTHF, a vital cofactor necessary for inosine monophosphate (IMP) biosynthesis. Purine metabolism, specifically the purine de novo biosynthesis pathway, plays a crucial role in the proliferation of cancer cells. Phosphoribosylaminoimidazole succinocarboxamide synthetase (PAICS), an enzyme involved in this pathway, is essential for DNA synthesis and has been linked to the progression and metastasis of cancer. PAICS, along with PPAT (phosphoribosyl pyrophosphate amidotransferase), another enzyme in the purine biosynthesis pathway, are thought to be key drivers in the metabolic reorientation of cancer cells towards aerobic glycolysis, commonly known as the Warburg Effect. This metabolic reprogramming redirects glycolytic intermediates from the tricarboxylic acid (TCA) cycle towards purine de novo biosynthesis, thereby facilitating cancer cell proliferation and invasion. Given these insights, our hypothesis is that the upregulation of DYRK3 induced by radiation exposure could potentially influence the purine biosynthetic pathway. This influence might specifically alter the glycolysis process involving PAICS and PPAT, thereby impacting the overall metabolic profile and behavior of cancer cells. This hypothesis underlines the intricate relationship between radiation exposure, DYRK3 expression, and metabolic pathways in cancer cells, particularly in the context of purine biosynthesis and energy production, offering new avenues for understanding and potentially targeting cancer cell proliferation and metastasis.

The primary objective of this research is to examine the impact of DYRK kinases on the development of radioresistance and metastasis in oral squamous cell carcinoma (OSCC). Existing evidence suggests that protein kinases play crucial roles in regulating essential cellular signals and diverse biological processes related to tumor cell growth, proliferation, differentiation, metabolism, apoptosis, and drug resistance/sensitivity [14]. The aberrant expression of DYRK kinases has been associated with various human diseases, including different types of cancer [15]. Radioresistance poses a significant obstacle in cancer treatment, especially in the context of OSCC. Furthermore, altered metabolism, a characteristic feature of cancer including OSCC, may independently contribute to the regulation of radioresistance in therapy. Nevertheless, the specific role of DYRK3, a vital member of the DYRK family, as a tumor suppressor or promoter, and its potential involvement in cancer, remain largely unexplored [16]. Additionally, the connection between DYRK3 and radioresistance in OSCC tumorigenesis requires further investigation. This study aims to shed light on the significance of DYRK kinases in OSCC by exploring their role in radioresistance and metastasis. Understanding the specific function of DYRK3 and its association with radioresistance could potentially contribute to the development of novel therapeutic approaches for OSCC treatment.

## 2. Results

### 2.1. Expression Levels of DYRK3 in the Whole Blood and Tumor Tissues of Patients with Oral Squamous Cell Carcinoma (OSCC)

To gain a comprehensive understanding of the role and impact of DYRK3 in the context of OSCC, it is imperative to study its expression levels both in the whole blood and the tumor tissue of the affected individuals. Investigating the expression levels in these two distinct environments can potentially furnish significant insights into the disease’s development, progression, and the possible effects of DYRK3 at different stages of OSCC. We gathered and analyzed leftover surgical tissues from 50 patients undergoing radiation therapy for Head-Neck Squamous Cell Carcinoma (HNSC) at Tri-Service General Hospital (TSGH). This was done to investigate the expression of upstream and downstream targets of DYRK3 in these samples. By analyzing whole blood samples, researchers can discern the systemic influences and manifestations of the disease, obtaining a broader perspective on how OSCC may be affecting the entire body. On the other hand, studying the expression directly within the tumor tissue allows for a more localized and precise understanding of how DYRK3 interacts and influences the tumor environment, cell growth, and proliferation. This approach, utilizing both blood and tumor tissue analysis, will enable a more nuanced comprehension of DYRK3’s role in oral squamous cell carcinoma and may contribute to the identification of novel diagnostic markers, prognostic indicators, or therapeutic targets for this particular type of cancer. We aim to explore the potential of utilizing these factors as biomarkers for precision treatment. Table 1 presents the relevant clinical data on DYRK3 expression and correlation analysis. In Figure 1, the varied expression levels of DYRK3 in the blood and tumor tissues of OSCC are highlighted. Figure 1A contrasts the mRNA expression levels in the whole blood from both healthy individuals and those diagnosed with OSCC, with results derived from qRT-PCR assessments. Figure 1B provides a visual representation of the disparity in DYRK3 expression between OSCC radio-resistance samples (*n* = 30) and adjacent normal tissues (*n* = 30). Furthermore, Figure 1C depicts the differential expression of markers Ki-67 and p53-mut in patients with OSCC, specifically comparing those with high and low levels of PAICS expression. To assess the protein expression of DYRK3 in patients with radiation-resistant OSCC, immunohistochemical (IHC) staining was performed (Figure 1D). Next, we investigated the influence of PAICS and DYRK3 expression after radiotherapy on the overall survival rate of head and neck cancer patients in the Cancer Genome Atlas (TCGA) cohort. The results demonstrated that PAICS and DYRK3 expression levels significantly impacted the overall survival outcome (Figure 1E). The regulatory function and expression levels of DYRK3 were found to have a positive correlation with the expression levels of Ki-67 and p53, thereby implying the potential use of immunohistochemical staining for Ki-67 and p53 as indicative markers for detecting early postoperative recurrence in OSCC cases. This initial set of data significantly reinforces our proposed theory, indicating a pivotal role of DYRK3 in the emergence of malignant phenotypes within OSCC. The intricate interplay and regulatory mechanisms of the PAICS/DYRK3 axis were observed to have a conspicuous positive correlation with the expression patterns of Ki-67 and p53.

### 2.2. Co-Expressed Genes Were Subjected to Weighted Gene Correlation Network Analysis (WGCNA)

In this study, our focus was to assess the expression and relationship between DYRK3 and PAICS in human tumor xenograft samples with varying degrees of radioresistance and radiosensitivity. The objective was to investigate the correlation between high DYRK3 expression, DYRK3/PAICS signaling, and sensitivity to radiation therapy. Firstly, we conducted an analysis of differentially expressed genes (DEGs) using the GEO data sets (GSE9712) and generated a heat map to visualize the expression patterns (Figure 2A). Additionally, volcano plots and heat maps were employed to identify numerous genes that exhibited significant differences between radioresistant and radiosensitive OSCC samples (Figure 2B,C). To explore the correlation between DYRK3 and PAICS mRNA levels, we calculated Pearson’s correlation coefficients, revealing a positive association, which indicated that DYRK3 expression is positively correlated with PAICS expression (Figure 2D). Furthermore, we examined the expression of the DYRK3/PAICS axis in different stages of head and neck cancer to gain insights into the molecular regulatory changes occurring during cancer progression. The findings indicated dynamic alterations in the regulation of the DYRK3/PAICS axis throughout the various stages of the disease. (Figure 2E). Next, we performed the screening of modules for co-expressed genes using the Weighted Gene Correlation Network Analysis (WGCNA) program, utilizing the GEO data set GSE9712. To begin, we constructed a WGCNA cluster dendrogram of the differentially expressed genes (DEGs) (Figure 2F). Each vertical line in the dendrogram represents a gene, and the color coding beneath the dendrogram indicates the assigned and merged modules generated by WGCNA. Furthermore, we visualized the mRNA network of the turquoise module using the Cytoscape software platform 3.10.1 (Figure 2G). The circles in the network represent key genes involved in this module. To gain deeper insights into the expression of the PAICS gene in tongue tissue, we consulted The Human Protein Atlas online database. This resource revealed the single-cell gene expression profile of PAICS in human tongue tissue, highlighting its notable presence in epithelial cells and keratinocytes, as illustrated in Figure 2H. Additionally, we assessed the mRNA expression levels of DYRK3, PAICS, and PPAT in clinical samples from the TCGA HNSC cohort, which is detailed in Figure 2I. This analysis offers crucial data about the expression of these genes in oral squamous cell carcinoma (OSCC).

### 2.3. Kinome siRNA (Small Interfering RNA) Library Screening for Kinases in OSCC Cells HN12 and SAS Cell Line Were Evaluated for Kinase Involvement in Radioresistance

The human kinome comprises around 538 kinase genes, encompassing 89 tyrosine kinases, 429 serine/threonine kinases, and 20 lipid kinases. These kinases are vital for signaling mechanisms that govern a range of cellular physiological activities, including cell proliferation, migration, and differentiation. Notably, tyrosine kinases, among others, are known to play a significant role in the development of tumors, malignancies, and resistance to cancer treatments. Consequently, these kinases have become focal points for targeted cancer therapies in clinical practices. In our study, we performed a comprehensive kinome siRNA (small interfering RNA) library screening to identify kinases that may play a role in radioresistance in OSCC cells. Specifically, we focused on evaluating the HN12 and SAS cell lines to determine the presence of these kinases. To begin the screening process, we seeded the HN12 and SAS (Figure 3A) cells and transfected them with the arrayed kinome siRNA library. In order to investigate the kinases that influence cell viability in oral squamous cell carcinoma (OSCC), we conducted a screening of human OSCC cells, specifically the HN12 and SAS cell lines, using an siRNA library that targets 709 human kinases and related genes. Human gingival fibroblasts (HGF-1) served as the control group for this study. From this screening, the top 15 gene targets were initially chosen based on their role in reducing cell proliferation in OSCC cells. Subsequent counter-screening with human gingival fibroblasts (HGF-1) highlighted the impact on several genes, including DYRK3, DDR1, WEE1, and CDC2L2. Focusing on these findings, DYRK3 was selected for further validation to assess both the effectiveness of its knockdown and its influence on cell viability.

Each gene in the library was targeted with pooled siRNAs (5 nM) for 72 h. To assess cell viability, we conducted an SRB assay. Following the initial screening, we identified the top-ranked siRNAs that demonstrated a significant reduction in cell viability for both HN12 and SAS cells. To validate the efficiency of the siRNA knockdown, we performed qRT-PCR analysis to measure the mRNA levels of the targeted genes, using scrambled siRNA as a control. The fold change in mRNA levels for each siRNA knockdown is indicated above the corresponding bar. Among the various genes that exhibited knockdown, we selected DYRK2 and DYRK3 for further investigation (Figure 3B). Notably, DYRK3 was specifically chosen for subsequent studies. To ensure that the selected genes do not impact normal physiological processes, we extended our analysis to include HGF-1 cells, which are normal human gingival cells. Our observations indicated that the knockdown of these selected genes resulted in reduced viability specifically in cancer cells, without affecting the viability of normal HGF-1 cells. This suggests that DYRK3 may be of importance and warrant further investigation. Finally, we used the GSE42743 data set of Gene Expression Omnibus (GEO) to analyze the expression profile of DYRK2 and DYRK3 genes in oral cancer and adjacent normal tissues. Gene expression profiling involves assessing the array of genes that are transcribed under certain conditions or within a particular cell type. This process provides a comprehensive overview of cellular functions by mapping out gene expression patterns (Figure 3C). 

Our study sheds light on the pivotal function of the DYRK3 gene, particularly its role as a crucial kinase in the context of oral squamous cell carcinoma (OSCC). This significance is observed across different cell types within OSCC, encompassing both radioresistant (HN12) and radiosensitive (SAS) cells. Intriguingly, we found that depleting DYRK3 not only affects the viability of radioresistant cell lines but also has a marked impact on radiosensitive cell lines. DYRK3 plays a multifaceted role in the survival and proliferation of OSCC cells, regardless of their radiation response characteristics. The fact that DYRK3 depletion impacts both types of cell lines hints at its overarching influence in the cellular processes of OSCC. By understanding the specific functions and impacts of genes like DYRK3, we can better tailor treatments to combat OSCC more efficiently, potentially improving patient outcomes in both radioresistant and radiosensitive scenarios.

### 2.4. DYRK3 Overexpression Enhances the OSCC Cells’ Proliferation, Migration, Invasion, and Self-Renewal Ability

Our studies indicate that the heightened presence of DYRK3 leads to a substantial increase in proliferation, migration, invasion, colony formation, and sphere formation in oral squamous cell carcinoma (OSCC) cells. We utilized the quantitative real-time polymerase chain reaction (qRT-PCR), a powerful technique known for its efficacy in determining gene expression, to measure DYRK3 levels following the introduction of the OE-DYRK3 (overexpression vector) into OSCC cells. This evaluation was performed across a variety of OSCC cells, specifically focusing on both radioresistant (HN12) and radiosensitive (SAS) variants (Figure 4A). Following the introduction of the OE-DYRK3 into oral squamous cell carcinoma (OSCC) cells, we conducted a comprehensive analysis to assess the impact on various protein levels and gene expressions. Specifically, we measured the levels of DYRK3 and PAICS proteins and monitored the expression of pluripotency markers, notably PPAT. Additionally, our focus extended to examining the presence of Glut-1, a gene known for its association with radioresistance in OSCC. Glut-1, or Glucose Transporter 1, is a critical marker when studying radioresistance in OSCC. Its heightened expression within these cells is a significant indicator of a poor response to radiation therapy. This is particularly important as increased Glut-1 levels are often correlated with a diminished effectiveness of radiation treatment and, consequently, a shorter survival period for patients affected by OSCC. Our investigation, as illustrated in Figure 4B, aimed to understand these relationships better and provide deeper insights into how these proteins and genes interact and influence each other, especially in the context of radioresistance in OSCC. This research is crucial in developing more effective treatment strategies for OSCC, particularly for cases where conventional radiation therapy may not be as effective due to the presence of radioresistant cell populations. By utilizing the Sulforhodamine B (SRB) assay, we assessed cell viability in OSCC cells where DYRK3 was overexpressed. This allowed us to accurately gauge cell growth and cytotoxicity over a period of 24–48 h (Figure 4C). We noticed significant morphological alterations in the OSCC cells following the transfection with the O-DYRK3. Such transformations in cells can often indicate the cellular response to a specific treatment or condition, serving as a gauge of cellular health or functionality (Figure 4D). After overexpressing DYRK3, we conducted several assays, including migration/wound healing, sphere formation, and migration (Figure 4E). Furthermore, our neuro sphere formation assays suggested that elevating DYRK3 levels fortified the stemness of OSCC cells, a quality closely associated with the self-renewal and differentiation capacities of cancer cells (Figure 4F). In conclusion, our findings provide compelling evidence that overexpressing DYRK3 in OSCC cells significantly enhances their oncogenic properties, especially in relation to colony formation, migration, invasion, and tumor sphere generation. In parallel, we also observed a notable rise in the expression of proteins, such as PAICS, Glut-1, and PPAT, further affirming the intricate role of DYRK3 in the regulation of OSCC progression.

### 2.5. Diminished Purinosome Formation Following DYRK3 Gene Knockdown in Two Radiation-Resistant OSCC Cell Lines

Figure 5 presents a comprehensive overview of our experimental approach for the investigation of the impact of reducing DYRK3 expression on the formation of purinosomes in OSCC IR cells. Ionizing radiation (IR)-resistant OSCC cells are generated as described in the Materials Methods section. To fine-tune our experimental parameters, we initially conducted an examination of cell line expression in HSC3 and SAS cells (Figure 5A). We utilized the DepMap online tool (https://depmap.org/, accessed on 1 November 2023). to identify vulnerabilities in cancer cells and potential targets for therapeutic advancements. Our findings confirmed the relevance of HSC3 and SAS cell lines with respect to PAICS and DYRK3 expression, validating their suitability for this study. To further elucidate the impact of DYRK3 depletion, we performed Western blot analysis (Figure 5B), examining the levels of DYRK3, PAICS, PPAT, and the radioresistance-associated gene (Glut-1) in shDYRK3-transfected OSCC cells. We also conducted functional assays to investigate the effects of DYRK3 depletion on OSCC IR cells. Next, we assessed DYRK3 transfection with shDYRK3 through qRT-PCR in OSCC IR cells HSC3 and SAS (Figure 5C) and analyzed gene expression changes following DYRK3 gene knockdown using qRT-PCR (Figure 5D). We also conducted functional assays to investigate the effects of DYRK3 depletion on OSCC IR cells. Wound healing and tumor sphere formation experiments (Figure 5E) demonstrated slower healing of scratch wounds and the suppression of OSCC cell stemness in response to DYRK3 depletion at both 24 and 48 h. Additionally, we evaluated cell viability in shDYRK3-inhibited OSCC cells using the SRB assay at 24–48 h (Figure 5F), revealing a decrease in cell viability. Lastly, we delved into the impact of suppressing the DYRK3 gene on the development of purinosomes in two specific oral squamous cell carcinoma (OSCC) cell lines known for their resistance to radiation, as depicted in Figure 5G. We employed cell fluorescence staining techniques to compare these effects. Our findings revealed that in the radiation-resistant HSC3 and SAS cell lines, there was a noticeable elevation in the expression levels of DYRK3. This observation implies that the formation of purinosomes might be a response to the acquisition of radioresistance, coupled with an increase in DYRK3 levels. Furthermore, when we reduced the expression of DYRK3 in these radiation-resistant cell lines, we observed a significant decrease in the formation of purinosomes. This led us to conclude that DYRK3 likely plays a crucial role in the assembly and regulation of purinosomes within these specific cell lines. Overall, our experiments provide a more comprehensive insight into the effects and functional implications of diminishing DYRK3 expression in OSCC cells that are resistant to ionizing radiation (IR). This research contributes to a broader understanding of the cellular mechanisms at play in cancer cells’ response to radiation therapy.

### 2.6. Significant Suppression of Metastasis in Patient-Derived Xenograft Mouse Models In Vivo Is Achieved through the Inhibition of DYRK3

We conducted an analysis to assess the remedial impacts of the DYRK3 inhibitor (GSK-626616) by constructing a patient-derived xenograft mouse model. This involved the orthotopic implantation of mice with tumor cells derived from patients with oral squamous cell carcinoma who had experienced a relapse post-radiotherapy. Through tissue immunostaining, we observed abnormal expression in the DYRK3/PAICS axis in these cells. For in vivo validation of our in vitro findings, we introduced the derived tumor cells into the right flank of female NOD-SCID mice. We divided the mice into four distinct groups: a control group, a group subjected to cisplatin only (administered orally five times weekly), a group exposed to the DYRK3 inhibitor only (given orally five times weekly), and, lastly, a group that received a combination of both treatments (Figure 6A). In mice treated with the DYRK3 inhibitor, there was a slight increase in tumor sizes at certain intervals, resulting in a 1.6 times larger size by the 8th week compared to the control group, a difference that was statistically significant (*p* < 0.01). Furthermore, the DYRK3 inhibitor increased the survival rate significantly, with no notable impact on the weight of the mice by the 6th week (Figure 6B). The Kaplan–Meier survival curve shows that the mice receiving combined treatment exhibited the highest survival ratio compared with the remaining mice (Figure 6C). Subsequent experiments using tumor samples from the xenograft mouse models revealed significant suppression of DYRK3, PAICS, and Ki-67 proteins in the mice treated with the DYRK3 inhibitor and the combined treatment, relative to the control mice. The Q-score of the tissue staining is also documented herein. The findings indicate a pivotal role of DYRK3 in the development and metastasis of OSCC, and its modulation of specific markers (Figure 6D). These outcomes highlight the crucial function of DYRK3 in the malignant advancement of OSCC and its effect on the modulation of specific markers.

## 3. Discussion

Radiotherapy is a widely used treatment option for advanced-stage cancer, with about half of all cancer patients receiving this alongside primary treatments like surgery and chemotherapy [17]. This method employs high-dose ionizing radiation to induce cell death by damaging cellular DNA. However, resistance to radiation, also known as radiation resistance, poses a significant clinical challenge for various cancer types including glioblastoma, breast cancer, and esophageal adenocarcinoma [18]. Certain organelles, especially mitochondria, play pivotal roles in influencing a tumor’s response to radiation as they regulate several processes tied to radioresistance [19]. Within the framework of purine metabolism, a key component known as the purinosome has been identified. The formation of the purinosome closely correlates with the cell cycle, presenting a new therapeutic opportunity for treating cancers by targeting purinosome formation and purine metabolism [20]. Recent research findings further underscore the importance of purine synthesis in the aggressive behavior exhibited by glioblastoma (GBM) and other cancer types [21]. Elevated rates of de novo purine and pyrimidine synthesis aid in maintaining glioma-initiating cells, potentially contributing to therapy resistance and recurrence in GBM. While cancer metabolism has attracted increasing attention in recent years, the link between metabolism and DNA damage/DNA repair in cancer still requires further investigation [22]. Thus, this report aims to shed light on the association between DNA repair, DNA damage, and purine metabolism.

PAICS, an enzyme involved in the metabolic pathway of purine nucleotides, is a key contributor to the proliferation and spread of tumors [23]. This dual-function enzyme, which hosts phosphoribosylaminoimidazole carboxylase activity in its N-terminal region and phosphoribosylaminoimidazole succinamide synthetase activity in its C-terminal region, mediates the sixth and seventh stages of purine biosynthesis [24]. Notably, PAICS is pivotal in pancreatic ductal adenocarcinoma (PDAC) and is thus regarded as a promising therapeutic target due to its susceptibility to small molecule targeting [25]. Despite its recent emergence as a potential target, its role in radiation-resistant oral cancer has not been extensively discussed. DYRK, standing for dual-specificity tyrosine-regulated kinase, is a protein kinase family conserved across multiple species. It has the capability to phosphorylate numerous proteins integral to a variety of cellular processes, including proliferation of cancer cells, enhanced drug resistance, and signaling pathways linked to tumor progression [13]. Abnormal regulation or expression of DYRK kinase is associated with diverse human diseases, including cancer. Dual-specificity tyrosine phosphorylation-regulated kinase 1A (DYRK1A), a protein kinase that is preserved across evolutionary scales, plays a crucial role in the onset of Down syndrome and other diseases. It influences neurodevelopment, cell proliferation and differentiation, tumor formation, and the pathogenesis of neurodegenerative diseases. Furthermore, DYRK1A is critical in the pathogenesis of various diseases and the regulation of signal pathways. Elevated levels of DYRK1B protein and mRNA in tissues of colon adenocarcinoma are essential for tumor initiation and progression, which emphasizes the importance of DYRK1B detection for prognosis purposes. Studies on cervical cancer revealed that the expression of DYRK1B protein increases in parallel with the progression of cervical lesions, showing a high expression in cervical cancer tissues and cells. Increased levels of DYRK2 in patients are correlated with better prognoses, improved responses to chemotherapy, and increased survival rates, particularly in instances of liver metastasis originating from colorectal cancer. In contrast, in ovarian serous adenocarcinoma, diminished DYRK2 expression is associated with a poorer prognosis [16].

In cases of hepatocellular carcinoma (HCC), DYRK3 expression was significantly lower compared to normal controls [26]. However, the introduction of DYRK3 in HCC cells led to a marked reduction in tumor growth and metastasis in xenograft tumor models. As such, deciphering the molecular mechanisms that control DYRK can offer fresh perspectives on tumor cell plasticity and responses to both traditional and new treatments. This understanding could pave the way for the identification of novel drug targets for a more efficient, less toxic, and personalized therapy approach for patients with radiation-resistant oral squamous cell carcinoma (OSCC).

The purine biosynthetic pathway has recently emerged as an essential provider of metabolic intermediates for cancer-related processes. There is a growing body of evidence pointing to the multifaceted roles of PICAS/DYRK in numerous cellular and physiological responses across various cell types. Despite this, our understanding of the regulation of purine metabolism in oral squamous cell carcinoma (OSCC) remains quite limited. DYRK1A, 1B, 2, and 3 belong to the dual-specificity tyrosine phosphorylation-regulated kinase (DYRK) family. Our research indicates that DYRK3 expression is notably higher in human cancer tissues than in healthy ones.

Our hypothesis posits that the cancer-inducing role of PAICS/DYRK3 could be instrumental in fostering radioresistance and metastasis in OSCC by boosting the functionality of the purinosome, which in turn enhances the stemness, tumorigenicity, and drug resistance of OSCC. However, a decrease in PAICS expression might potentially counteract these effects. We propose a new PAICS/DYRK3 signaling pathway that could activate purinosome formation reprogramming, which may serve as a targeted treatment for radioresistant OSCC. Purine metabolism is a critical process for tumor development, with PAICS able to trigger cancer purine metabolism and thus facilitate tumor progression. The results obtained from our experimental investigations have provided compelling evidence supporting the critical involvement of DYRK3 and PAICS in oral squamous cell carcinoma (OSCC). Notably, both OSCC patients and cell lines demonstrated a pronounced overexpression of DYRK3, which exhibited a strong correlation with PAICS expression (Table 1, Figure 1 and Figure 2). These findings further substantiate the significance of DYRK3 and PAICS in OSCC biology. Importantly, the upregulation of DYRK3 and its correlation with PAICS expression were found to be associated with a poor prognosis in OSCC patients (Figure 1). This observation underscores the potential clinical relevance of DYRK3 and PAICS as prognostic markers in OSCC. Moreover, our findings shed light on the proposed involvement of the PAICS/DYRK3 axis in purinosome formation, specifically in the context of radioresistance and metastasis development in OSCC (Figure 5). This novel insight suggests that the dysregulation of the PAICS/DYRK3 axis may contribute to the acquisition of radioresistance and the metastatic potential of oral squamous cell carcinoma. These results emphasize the potential significance of the PAICS/DYRK3 axis in orchestrating the modulation of treatment effectors, thus affecting the therapeutic outcomes in OSCC. Additionally, the dysregulation of this axis appears to play a complex role in tumorigenicity, disease progression, and the evasion of therapy in patients with oral squamous cell carcinoma. Lastly, we suggest that reducing PAICS expression could lessen the invasion and migration capabilities of radioresistant OSCC, and combining this with cisplatin treatment could enhance apoptosis in radioresistant OSCC (Figure 3 and Figure 4). Taken together, our findings highlight the crucial involvement of DYRK3 and PAICS in OSCC pathogenesis. This knowledge opens new avenues for further research on the PAICS/DYRK3 axis as a potential therapeutic target and may ultimately contribute to the development of novel treatment strategies for patients with oral squamous cell carcinoma. Indeed, when cancer cells were exposed to a dual treatment comprising methotrexate, a well-established inhibitor of purine biosynthesis, and an Hsp90 inhibitor, it resulted in a synergistic cytotoxic impact [27]. While numerous kinases have been identified as critical players in cancer malignancy management, the specific role of kinases in promoting radioresistance in oral squamous cell carcinoma (OSCC) remains underexplored. In our study, we conducted a targeted kinome analysis for OSCC and pinpointed DYRK3 as a key factor in fostering radioresistance. We theorize that radiation-triggered upregulation of DYRK3 impacts the dynamics of purinosome formation, leading to metabolic shifts and heightened invasiveness in OSCC post-radiation treatment. Considering these roles of DYRK3, we suggest that it could serve as a potential therapeutic target to inhibit the progression of OSCC following radiotherapy. This research confirms the potential of the purinosome as a promising target for innovative cancer chemotherapy and introduces the prospect of creating novel combination therapies that hinder purine biosynthesis by targeting both specific enzymes within the pathway and the formation of the purinosome complex.

## 4. Materials and Methods

### 4.1. Collection of Clinical Specimens and Creation of In Vitro Cell Model of Patient-Derived Primary Cells

Obtain tissue samples from 50 individuals with radiation therapy oral squamous cell carcinoma (OSCC) at Tri-Service General Hospital (TSGH). The research, which focuses on studying the expression of DYRK3/PAICS upstream and downstream targets, commences after receiving approval from the institutional review board (IRB: N202304008) of Taipei Medical University and adhering to the ethical guidelines outlined in the Declaration of Helsinki for biomedical research. To assess the expression of DYRK3 in recurrent OSCC, we construct a tissue array comprising samples from a total of 50 subjects. This array will consist of 20 non-recurrence tissues and 30 radio-resistance tissues obtained from patients with radioresistant and metastatic OSCC. Immunohistochemical (IHC) staining will be conducted on the tissue array using an antibody specifically targeting DYRK3 (at a concentration of 1:400, ab96617, Abcam, Greater Boston, MA, USA). The staining process will follow the standard IHC protocol, including the use of mouse immunoglobulin G (IgG) as a negative control at a comparable dilution. To establish primary cells derived from the patients’ tumors, small pieces of tumor tissue (measuring 2 mm^3^) will be surgically obtained during the procedure. These tissue fragments, known as explants, will be placed in a 24-well plate coated with fetal bovine serum. Essential nutrients will be supplemented to facilitate optimal growth. The medium volume will be adjusted to ensure that the explants adhere to the bottom of the plate and do not float. Regular monitoring of the explants will take place, and any explants exhibiting fibroblast growth will be discarded, retaining only those with a single layer of epithelium. Once the epithelial layer forms a distinct halo, the explants will be transferred to a new Petri dish for further passages. This continuous process of passage will be repeated to acquire primary tumor cell lines.

### 4.2. Human Cell Lines of Oral Squamous Cell Carcinoma (OSCC) Resistant to Ionizing Radiation (IR)

To create cell lines that are resistant to radiation, we followed a previously published study. The SAS and HSC3 cell lines used in this experiment were obtained from the American Type Culture Collection (ATCC, Rockville, MD, USA). The SAS cell line, derived from a tongue squamous cell carcinoma in a Japanese patient, is an established oral cancer cell line frequently utilized in oral cancer research. These SAS oral cancer cells possess mutations in the p53 and p16 genes, which are mutations that are typically found in oral squamous cell carcinomas. Similarly, the HSC-3 cell line, originating from human tongue squamous carcinoma, serves as an effective model for investigating metastatic squamous cell carcinoma. To induce the production of radiation-tolerant cells and determine the optimal conditions for irradiation, the SAS and HN12 cell lines were exposed to radiation doses ranging from 2 to 10 Gy for 5 consecutive days. By conducting this procedure, we determined the maximum tolerated dose (MTD). We utilized a precision X-ray irradiator (North Branford, CT, USA) to irradiate the cells using X-rays. Cells from oral squamous cell carcinoma (OSCC) that survived after undergoing 30 cycles of irradiation, resulting in a cumulative dose of 60 Gy, were classified as radiation-resistant. The medium for the SAS and HSC3 cell lines was changed every 48 h, and the cells were assessed using a cell function assay.

### 4.3. shRNA-Mediated DYRK3 Knockdown and Overexpression

We employed a lentiviral technique to reduce DYRK3 expression in SAS and HSC3 cells. Specifically, an shRNA construct was acquired from Origene (Cat#: TL309236V, Rockville, MD, USA) for this purpose. To create lentiviral particles carrying DYRK3-targeting constructs or scrambled controls, HEK293T cells were transfected with Lipofectamine 3000 (Thermo Fisher Scientific, Waltham, MA, USA). The resulting viral supernatant was then utilized to infect SAS or HSC3 cells in the presence of polybrene (4 μg/mL; Sigma-Aldrich, St. Louis, MO, USA). Subsequently, the transduced cells were cultured in a medium supplemented with puromycin (3 μg/mL, Sigma-Aldrich, St. Louis, MO, USA). We utilized data from the pcDNA3.1 mammalian expression vector (V79020, Invitrogen, Waltham, MA, USA) to formulate polymerase chain reaction (PCR) primers. Appendix A provides the vector map and primer sequences. Ten micrograms of either empty plasmid (pcDNA3.1 vector control plasmid DNA) or the DYRK3 expression plasmid (pcDNA3.1-CMV-DYRK3) were employed. The DNA-lipofectamine reagent complexes were allowed to incubate at room temperature for 30 min. Subsequently, this mixture was introduced into the well, and gentle agitation was achieved by rocking the plate back and forth. There was no need to remove the reagent complexes after transfection. The cells were then incubated at 37 °C in a CO_2_ incubator for 48 h. Real-time quantitative polymerase chain reaction (RT-PCR), along with quantitative Western blotting and detection, are essential techniques that can be employed to verify the impact of gene manipulation, such as knockdown or overexpression. RT-PCR is a highly sensitive and precise method that measures the amount of specific RNA, thus allowing for the quantification of gene expression levels following knockdown or overexpression. This is particularly useful in understanding the functional role of specific genes. On the other hand, quantitative Western blotting provides a robust approach to detect and quantify specific proteins, thereby confirming the effects of gene manipulation at the protein level. This combination of RT-PCR and Western blotting offers a comprehensive approach to validate the efficiency and functional consequences of gene knockdown or overexpression in various biological and clinical research contexts. Successful knockdown or overexpression of cells was confirmed through either quantitative reverse transcription polymerase chain reaction (qRT-PCR) or Western blotting. Appendix A provide the list of Q-PCR primers and antibodies.

### 4.4. Transient Transfection and Live Cell Fluorescence Imaging of OSCC-IR Cells

HSC3-IR and SAS-IR cells were produced following the previously described method and cultured for one week in purine-depleted medium, as outlined in prior studies. The fluorescent protein fusion vector PAICS-EGFP-N1 and PPAT-EGFP-N1 were modified from pEGFP-N1 (Takara Bio, Kusatsu shi, Shiga ken, Japan). Plasmids were constructed using established procedures [28]. All the gene inserts were verified through DNA sequencing, and the plasmids were subsequently introduced into competent cells XL1-Blue or DH5α. The vector map and primer sequences are presented in Appendix A. Afterward, they were isolated using the QIAprep Spin Miniprep Kit (Qiagen, Rockville, MD, USA). Subsequently, HSC3-IR and SAS-IR cells, which were cultivated in purine-depleted media, were transiently transfected with plasmid DNA at a concentration of at least 1 μg/μL, employing Lipofectamine 2000 according to the manufacturer’s guidelines. For live cell fluorescence imaging, cells were observed under physiologically relevant conditions (approximately 37 °C and 5% CO_2_) within a cell incubation chamber provided by Tokai Hit. A 60× Nikon Apo TIRF oil immersion objective, along with an S484/15x excitation filter and an S517/30m emission filter for GFP (Chroma Technology, Bellows Falls, VT, USA), was used for imaging. The de novo purine biosynthetic pathway is the predominant pathway in G1 phase when the purine demand for the cells is at its highest [29]. To synchronize the cells at the G1 phase, we introduced 0.5 mM of DB-cAMP from Sigma-Aldrich into the cell culture medium 5 h after the transfection of PACIS-GFP. This mixture was then incubated for an additional 18 h. Before imaging, the medium containing DB-cAMP was replaced with purine-depleted medium devoid of phenol red.

### 4.5. Ionizing Radiation (IR) Resistance Cell Functional Analysis and Gene Expression Evaluation

To generate tumor spheroids, SAS and HN12 OSCC cells were cultured for a period of 14 days, followed by a subsequent evaluation. The spheroids were visually counted under a microscope, and the efficiency of tumor sphere formation was calculated by determining the ratio between the number of spheres and the initially implanted cells. Migration distances were measured based on images taken at specific time points, capturing three random areas. Analysis of gap size was conducted using Image J 1.42 software. For cell invasion assays, three separate and random fields per well were photographed, and the cell count per field was quantified. The average value of the three measurements was obtained for each chamber, with at least three repetitions performed for each invasion detection. Total RNA was extracted from the samples (including tumor blocks and cell lines) using the TRIzol reagent from Life Technologies (Carlsbad, CA, USA), and its concentration was determined using a NanoDrop instrument from Thermo Fisher Scientific (Waltham, MA, USA). Quantitative real-time polymerase chain reaction (qRT-PCR) was carried out using the SYBR Premix Ex Taq II kit from Takara (Taipei, Taiwan) on a PCR detection system provided by Bio-Rad (Hercules, CA, USA).

### 4.6. Assessment of Tumor Spheroid Formation

In order to stimulate the formation of tumor spheres, OSCC cancer cells were placed onto serum-free low-adhesion culture plates containing Dulbecco’s modified Eagle’s medium (DMEM)/F-12 supplemented with N2 supplement (Invitrogen, Waltham, MA, USA), as well as 20 ng/mL of epidermal growth factor (EGF) and 20 ng/mL of basic fibroblast growth factor (bFGF) (known as stem cell medium; PeproTech, Rocky Hill, NJ, USA). Following a 2-week incubation period, the resulting spheres were enumerated under a microscope. The efficiency of tumor ball formation was determined by calculating the ratio between the number of tumor spheres and the initially seeded cells.

### 4.7. Migration Assays 

OSCC cancer cells were initially seeded and cultured in 6-well plates for a period of 24 h. To inhibit cell division, the cells were then treated with mitomycin (10 μg/mL) for 1 h. Subsequently, a linear scratch was created across the cell monolayer using a 200 μL pipette tip. After carefully removing any cellular debris, the cells were allowed to migrate over a duration of 24–48 h. The healing progress of the scratch was assessed by comparing micrographs taken before and after the scratch using a Nikon microscope. The migration distance was measured based on images obtained at specific time points from 3 randomly chosen fields. The size of the gap was analyzed using Image J 1.42 software. 

### 4.8. RNA Preparations

To obtain total RNA from tumor chunks and cell lines, TRIzol Reagent (Life Technologies, Carlsbad, CA, USA) was utilized, and the resulting RNA was quantified using NanoDrop (Thermo Fisher Scientific, Waltham, MA, USA). For the isolation of RNA from cell, a miRNeasy Micro Kit (QIAGEN, Hilden, Germany) was employed. In brief, 20 µL of cell suspension was mixed with 700 µL of QIAzol lysis buffer, and the subsequent steps were carried out following the vendor’s provided protocol. The RNA samples were then eluted using 25 μL of RNase-free water, and this elution step was repeated twice with the same volume of RNase-free water to concentrate the samples. The concentration of RNA in the samples was once again determined using NanoDrop.

### 4.9. Quantitative Real-Time PCR (RT-qPCR)

To quantify the levels of miRNA, we performed reverse transcription of miRNAs followed by RT-qPCR using the miDETECT A Track miRNA qRT-PCR Starter Kit (RIBOBIO, Guangzhou, China). For analyzing gene expression associated with the ACSL6/FASN axis, we synthesized first-strand cDNA using the PrimeScript 1st Strand cDNA Synthesis Kit (TaKaRa, Taipei, Taiwan). Subsequently, qRT-PCR was carried out using the SYBR Premix Ex Taq II kit (Takara, Taipei, Taiwan) on the CFX96 Real-Time PCR Detection System (Bio-Rad, Hercules, CA, USA).

### 4.10. Western Blotting

To determine the protein levels and assess the expression of related message transfer proteins, Western blotting was performed as follows. Initially, 10%, 12.5%, or 15% SDS-PAGE electrophoresis gels were prepared and placed into an electrophoresis tank filled with electrophoresis buffer. A total of 16 μL of the protein sample (20 μg total protein) was mixed with 4 μL of loading buffer, cooled, and denatured at 100 °C for 10 min. The denatured sample was loaded onto the prepared gel, and electrophoresis separation was conducted at 100 V for approximately 3 h. After electrophoresis, the gel was removed, and protein transfer was performed. The gel was placed in an ice-cold transfer buffer, and a pre-soaked PVDF membrane was placed on top of the gel within the transfer holder. The transfer process was carried out for 1 h. Subsequently, the PVDF membrane was taken out and incubated in a blocking buffer for 1 h and gently shaken at room temperature. For primary antibody incubation, the PVDF membrane was immersed in the TBST buffer containing the primary antibody and incubated overnight at 4 °C or for 2 h at 37 °C. Following primary antibody incubation, the membrane was washed three times with a washing buffer (TBS + 0.05% Tween 20) for 10 min each time. Next, the secondary antibody was added to the TBST buffer, and the membrane was allowed to react at room temperature for one hour. Subsequently, the membrane was washed three times with a washing buffer for 10 min each time. Finally, the membrane was developed using an ECL luminescence system, and the resulting bands were quantified using a densitometer (UVP/Bio Spectrum 810. GE, Boston, MA, USA).

### 4.11. Mouse Models Utilizing Patient-Derived Xenografts

To establish xenograft models, we employed mice that were homozygous for the severe combined immune deficient (SCID) mutation. We procured twenty-five 8-week-old female mice of the nonobese diabetic (NOD) and SCID strains from BioLASCO (Taipei, Taiwan). These mice were raised under controlled pathogen-free conditions, adhering to the guidelines outlined by the Animal Care and Use Committee of Taipei Medical University Animal Care and User Committee, with the Animal Use Agreement Ratification Affidavit number of Taipei Medical University LAC-2022-0467. In brief, radioresistant tumors from patients with oral squamous cell carcinoma (OSCC) were placed in RPMI 1640 on ice at the surgical site. The tumors were thinly sliced into 2–3 mm^3^ pieces and rinsed thrice with RPMI 1640. Subsequently, these samples were finely minced into fragments small enough to pass through an 18-gauge needle. They were then mixed in a 1:1 ratio (*v*/*v*) with Matrigel (obtained from Collaborative Research, Bedford, MA, USA), resulting in a total volume of 0.2 mL for each injection. This tissue mixture was subcutaneously injected into both flanks of 8-week-old male SCID mice. For the experimental groups, we obtained twenty 8-week-old female NOD/SCID mice from BioLASCO Taiwan and maintained them in the same controlled pathogen-free conditions. These mice were divided into four groups—vehicle, cisplatin, DYRK3 inhibitor (GSK-626616), or combination therapy—with each group consisting of five mice. The mice received different treatments: vehicle (PBS orally five times per week) or GSK-626616 (5 mg/kg, orally five times per week). The compound GSK-626616 is a specific inhibitor targeting the dual-specificity tyrosine-regulated kinase 3 (DYRK3), demonstrating potent inhibitory properties with an IC50 value of 0.7 nM. This particular inhibitor is utilized extensively in animal-based experimental research, particularly focusing on tumor treatment. Additionally, GSK-626616 plays a critical role in advancing the understanding of molecular interactions and mechanisms in cancer biology. It is used to examine the alterations in the network of proteins that interact with DYRK3 when exposed to the GSK-626616 inhibitor. This research is pivotal in comprehending how the inhibition of DYRK3 affects cellular processes and contributes to the development of targeted therapies for cancer treatment. Cisplatin, a platinum-containing anti-cancer agent, primarily acts by obstructing DNA synthesis and thereby inhibiting tumor cell growth. Employed either as a standalone treatment or in conjunction with other medications, it is effective in managing various forms of cancer. In our study, we established animal experimental groups for both single and combined treatment approaches to evaluate the drug’s therapeutic capabilities, focusing particularly on the effects of inhibiting the DYRK3 pathway.

To monitor tumor growth, we measured tumor volume using a standard caliper every other week, applying the formula V = (L × W^2^)/2, where ‘L’ represents the long axis of the tumor and ‘W’ signifies the width of the tumor. Following the conclusion of the experiments, humane euthanasia was performed on the animals, and tumor and tissue samples were collected for subsequent analyses.

### 4.12. Immunohistochemical Staining

Tissues obtained from experimental animals following sacrifice were fixed in 10% (*vol*/*vol*) formalin for 24 h and subsequently embedded in paraffin. In the case of bones, decalcification was performed prior to paraffin embedding. The paraffin-embedded tissue sections, with a thickness of 4 μm, underwent a series of steps: de-waxing by incubation with xylene for 2 min twice; rehydration with 100% ethanol for 2 min twice; 95% ethanol for 2 min, 75% ethanol for 2 min; and ddH_2_O for 2 min. The tissue sections were then stained with hematoxylin for 2 min, washed with tap water for 10 min, counterstained with eosin for 30 s, and subsequently dehydrated with 75% ethanol for 30 s twice with 95% ethanol for 30 s and 100% ethanol and xylene for 30 s each. After mounting the sections with a mounting medium, the stained tissue sections were examined using a light microscope, and the tumor areas were captured through digital photography. To calculate the tumor areas, Image J version 1.5 software (Wayne Rasband National Institutes of Health, Bethesda, MD, USA) was employed.

### 4.13. Statistical Analysis

Statistical analysis was performed using SPSS version 13.0 (SPSS Inc., Chicago, IL, USA). Each experiment was repeated three times, and the data in the figures were presented as mean ± standard deviation (SD). To compare two groups, a two-sided *t*-test was utilized. Statistical significance was defined as *p* < 0.05. The paired *t*-test or Wilcoxon signed-rank test was applied for continuous data analysis, while categorical data were examined using the χ^2^ test or Fisher’s exact test. Survival analysis was conducted using the Kaplan–Meier method, and differences between the curves were evaluated using the log-rank test.

## 5. Conclusions

In conclusion, as shown in the pictorial abstract of Figure 7, our experimental results highlight the pivotal roles of DYRK3 and PAICS in oral squamous cell carcinoma. The observed overexpression of DYRK3 and its correlation with PAICS, along with their association with a poor prognosis, underscore their importance in OSCC. Additionally, the proposed involvement of the PAICS/DYRK3 axis in purinosome formation sheds light on its role in the development of radioresistance and metastasis. This dysregulation of the PAICS/DYRK3 axis serves as a key modulator, influencing treatment effectors and contributing to tumorigenicity, disease progression, and therapy evasion in patients with oral squamous cell carcinoma. In this research, we explored new kinase-driven metabolic changes that contribute to the development of radioresistance and increased malignancy in oral squamous cell carcinoma (OSCC). Our findings indicate that radiation exposure triggers an increase in DYRK3 expression in OSCC cells. Furthermore, we observed that suppressing DYRK3 reduces the expression of several genes, including PPAT, PAICS, GIT2, ATM, and EGFR, which are linked to purinosome formation and metabolism. Notably, actively reducing DYRK3 expression following radiation exposure markedly decreases the migration and invasion capabilities of OSCC cells. Based on these findings, we propose that targeting DYRK3 could emerge as a novel approach to counteract OSCC malignancies by influencing purinosome formation and metabolism.

## Figures and Tables

**Figure 1 ijms-24-17346-f001:**
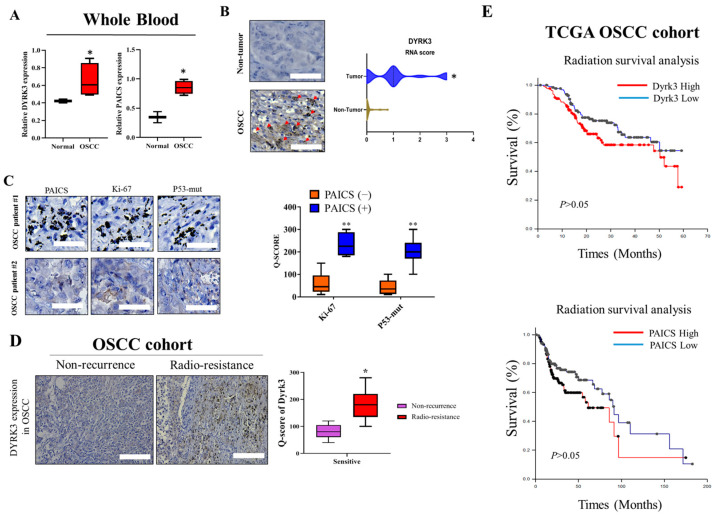
Expression Levels of DYRK3 in whole blood and tumor tissues of patients with OSCC. (**A**) DYRK3 mRNA expression in whole blood. Comparison of DYRK3 mRNA expression levels in whole blood between healthy individuals and patients with OSCC, assessed using qRT-PCR. (**B**) DYRK3 expression in tumor tissues. Visualization of DYRK3 expression differences between OSCC tumor samples (*n* = 50) and adjacent normal tissues (*n* = 50). Red arrows indicate the location of protein expression in cells after staining; (**C**) Ki-67 and p53-mut expression in high PAICS expression OSCC; (**D**) DYRK3 protein expression in radiation-resistant OSCC. Immunohistochemical (IHC) staining demonstrates DYRK3 protein expression in patients with radiation-resistant OSCC; (**E**) impact on overall survival. Investigation of the influence of PAICS and DYRK3 expression after radiotherapy on the overall survival rate of patients with oral squamous cell carcinoma in the Cancer Genome Atlas (TCGA) cohort. These findings underscore DYRK3’s pivotal role in the development of malignant phenotypes within OSCC and highlight the intricate interplay of the PAICS/DYRK3 axis with Ki-67 and p53 expression patterns. (* *p* < 0.05, ** *p* < 0.01.) Scale bar: 100 μm.

**Figure 2 ijms-24-17346-f002:**
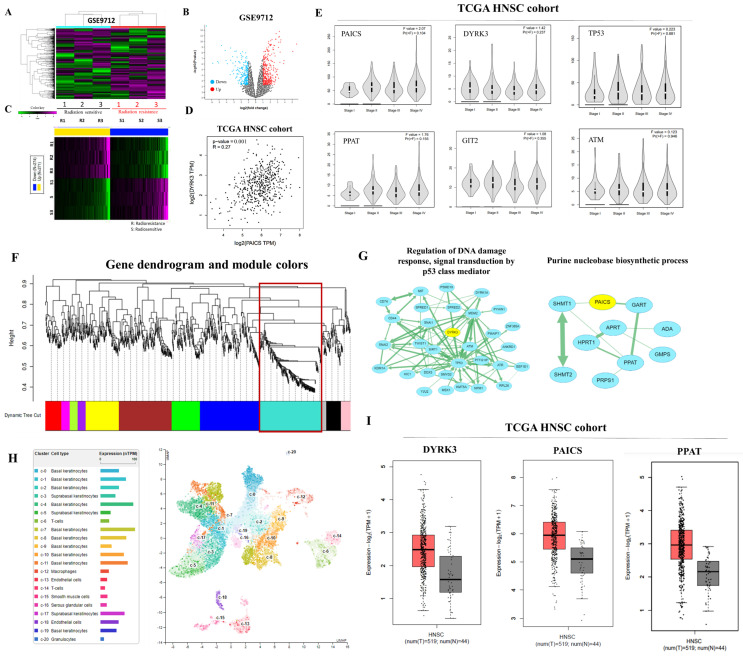
Expression and regulatory analysis of DYRK3 and PAICS in OSCC. (**A**) Differential gene expression analysis. Using GEO dataset GSE9712, a heatmap was generated to visualize expression patterns of differentially expressed genes (DEGs). (**B**) Identification of significant genes. Volcano plots and heatmaps were utilized to identify genes exhibiting significant differences between radioresistant and radiosensitive OSCC samples. (**C**) Heat map for gene differences. Heatmap illustrating the differential expression patterns of identified genes. (**D**) Correlation analysis. Pearson’s correlation coefficients were calculated to explore the positive association between DYRK3 and PAICS mRNA levels. (**E**) Molecular changes in cancer progression. Examination of the expression of the PAICS/ DYRK3 axis in different stages of head and neck cancer, revealing dynamic regulatory changes during cancer progression. (**F**) Co-expressed gene modules. Weighted Gene Correlation Network Analysis (WGCNA, Python 3.12.1) was performed on DEGs from GEO dataset GSE9712, resulting in a cluster dendrogram and module assignment. The red box is the target module. (**G**) mRNA network visualization. Visualization of the mRNA network of the turquoise module using Cytoscape, highlighting key genes in this module. (**H**): https://www.proteinatlas.org/ENSG00000128050-PAICS/single+cell+type/tongue, accessed on 1 November 2023. (**I**) Expression of clinical samples. Analysis of DYRK3, PAICS, and PPAT mRNA expression levels in clinical samples from TCGA HNSC cohort.

**Figure 3 ijms-24-17346-f003:**
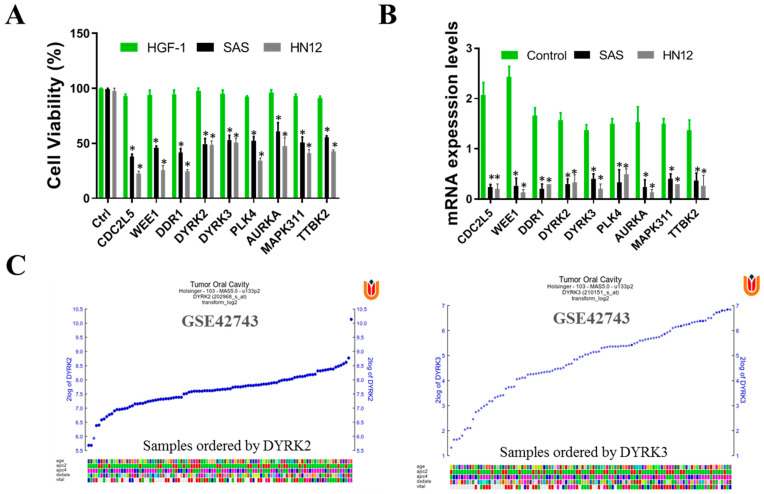
Kinome siRNA (small interfering RNA) library screening for kinases in OSCC cells. HN12 and SAS cell lines were evaluated for kinase involvement in radioresistance. (**A**) HN12 and SAS were seeded and transfected with the arrayed kinome siRNA library (5 nM pooled siRNAs for each gene) for 72 h. Cell viability was measured with SRB assay. Cells were transfected with the top-ranked siRNAs that reduced cell viability in both the cells. The siRNA was further assayed with HGF-1 (normal human gingival cells). To eliminate gene knockdowns that are also required for normal physiology, we observed that the viability of these selected genes reduced the cancer cell viability without affecting the cell viability of normal HGF-1 cells, suggesting the effect and importance of DYRK3 and DYRK2 may be worthy for follow up. (**B**) The mRNA levels of each gene were measured with qRT-PCR analysis to confirm the knockdown efficiency of siRNA using scrambled siRNA as a control. The data are represented in the fold change of mRNA level in each siRNA knockdown indicated above each bar. Out of many key gene knockdowns, DYRK2 and DYRK3 were selected, of which DYRK3 will be used for further study. (**C**) Expression profile of DYRK2 and DYRK3 genes in OSCC, GEO data GSE42743. The color blocks represent the visualization of the values in the data set. (* *p* < 0.05).

**Figure 4 ijms-24-17346-f004:**
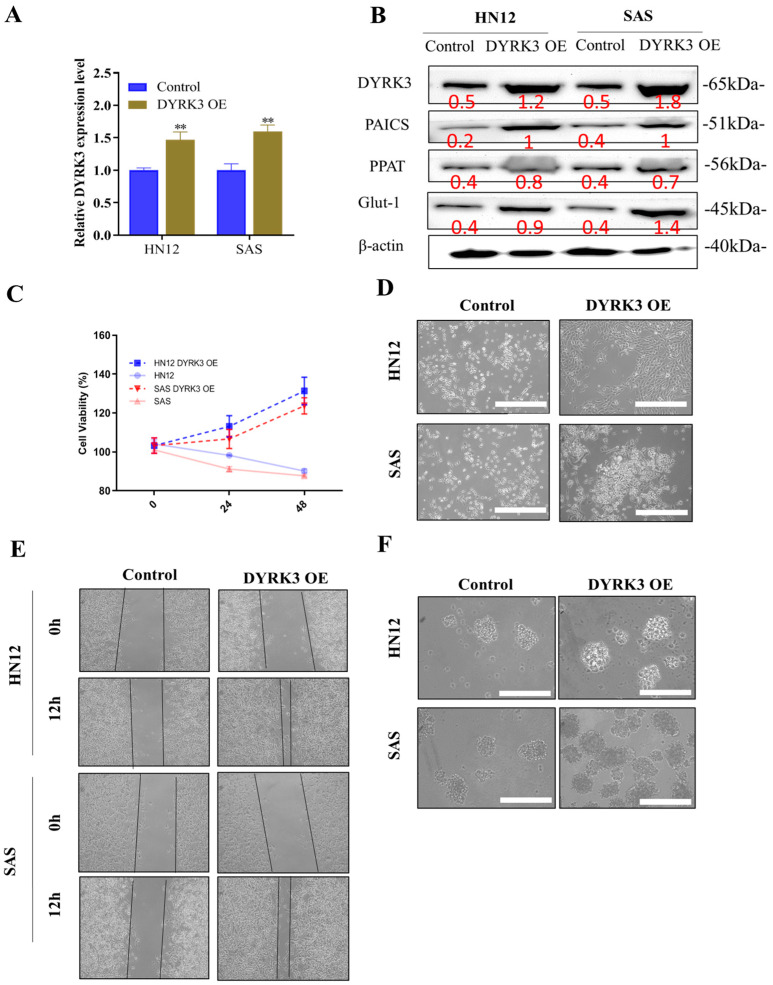
Impact of DYRK3 overexpression on OSCC cell behavior. (**A**) DYRK3 mRNA expression. Quantitative real-time polymerase chain reaction (qRT-PCR) analysis of DYRK3 mRNA levels in OSCC cells following overexpression with the OE-DYRK3 vector. Evaluation performed in both radioresistant (HN12) and radiosensitive (SAS) OSCC cell variants. (**B**) Protein expression and pluripotency markers. Western blot analysis depicting protein levels of DYRK3, PAICS, and pluripotency marker (PPAT) following OE-DYRK3 introduction. Evaluation of the radioresistance-associated gene Glut-1. (**C**) Cell viability assessment. Sulforhodamine B (SRB) assay was used to assess cell viability in OSCC cells overexpressing DYRK3, measuring cell growth and cytotoxicity over 24–48 h. (**D**) Morphological alterations. Observation of significant morphological changes in OSCC cells post-transfection with OE-DYRK3, indicative of cellular response to treatment or condition. (**E**) Migration/wound healing in OSCC cells after DYRK3 overexpression. (**F**) Sphere formation. Sphere formation assays indicating that elevated DYRK3 levels enhance the stemness of OSCC cells, a characteristic associated with self-renewal and differentiation capacities in cancer cells. (** *p* < 0.01). Scale bar: 5 μm.

**Figure 5 ijms-24-17346-f005:**
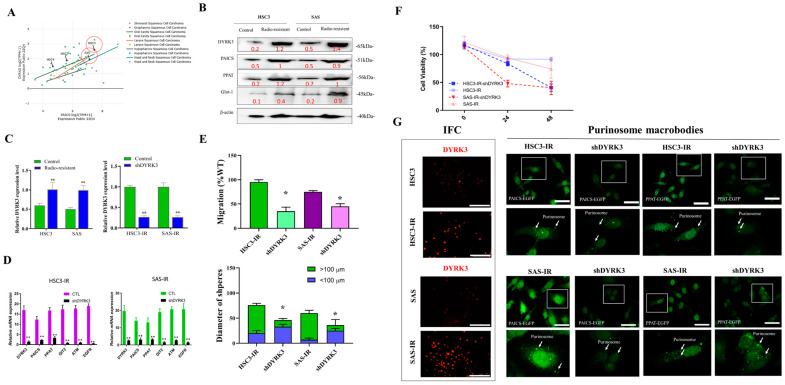
The impact of reducing DYRK3 expression on the formation of purinosomes in OSCC IR cells. (**A**) Examination of cell line expression to fine-tune experimental parameters in HSC3 and SAS cells. We leveraged the DepMap online tool (https://depmap.org/, accessed on 1 November 2023) to probe vulnerabilities in cancer cells and pinpoint potential targets for therapeutic advancements. Our results underscore the relevance of HSC3 and SAS cell lines concerning PAICS and DYRK3 expression, affirming their appropriateness for this investigation. (**B**) Western blot analysis of DYRK3, PAICS, PPAT, and the radioresistance-associated gene (Glut-1) in shDYRK3-transfected OSCC cells. (**C**) Assessment of DYRK3 transfection with shDYRK3 through qRT-PCR in OSCC IR cells HSC3 and SAS. (**D**) Analysis of gene expression changes following DYRK3 gene knockdown through qRT-PCR. (**E**) Wound healing and tumor sphere formation experiments revealing slower healing of scratch wounds and suppression of OSCC cell stemness in response to DYRK3 depletion at 24 and 48 h. (**F**) Evaluation of cell viability in shDYRK3-inhibited OSCC cells via the SRB assay at 24–48 h. (**G**) Diminished purinosome formation following DYRK3 gene knockdown in two radiation-resistant OSCC cell lines. The white arrow indicates the purinosomes’ location within the cell, and the dotted squares magnify the image of a single cell. The font color represents the fluorescent signal color. (* *p* < 0.05, ** *p* < 0.01). Scale bar: 5 μm.

**Figure 6 ijms-24-17346-f006:**
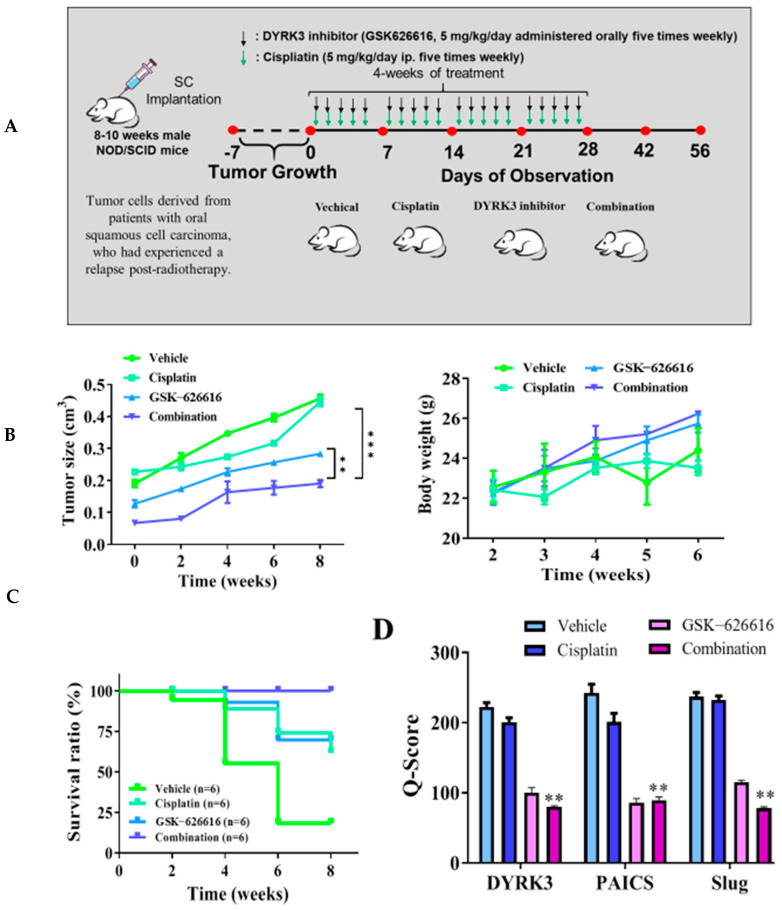
Inhibition of DYRK3 significantly suppresses metastasis in the patient-derived xenograft mouse models in vivo. (**A**) The flow chart for in vivo experimental design and treatment schedule. (**B**) The tumor size and body weight curve over time shows that cisplatin and the DYRK3 inhibitor in combination suppressed tumor growth and shows no apparent systematic toxicity in the mice receiving combined treatment. (**C**) The Kaplan–Meier survival curve shows that the mice receiving combined treatment exhibited the highest survival ratio compared with the remaining mice. (**D**) Immunostaining analysis of tumor sections showed that the combined treatment most prominently suppressed DYRK3, PAICS, Slug, and Ki-67 expression compared with other sections. (** *p* < 0.01, *** *p* < 0.001).

**Figure 7 ijms-24-17346-f007:**
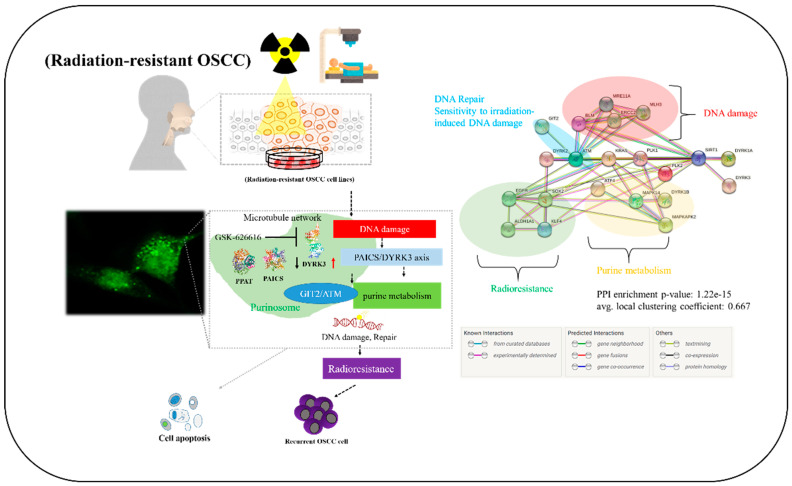
Visual summary. Our experimental results provide insights into the critical roles played by DYRK3 and PAICS in oral squamous cell carcinoma (OSCC). The documented overexpression of DYRK3, its association with PAICS, and their link to an unfavorable prognosis collectively emphasize their importance in OSCC. Moreover, the suggested involvement of the PAICS/DYRK3 axis in purinosome formation sheds light on its contribution to the development of radioresistance and metastasis. This disruption of the PAICS/DYRK3 axis serves as a pivotal regulator, impacting treatment outcomes and contributing to tumorigenesis, disease progression, and therapy resistance in patients afflicted with oral squamous cell carcinoma. Up and down arrows represent genes that are overexpressed or downregulated.

**Table 1 ijms-24-17346-t001:** Correlation between Dyrk3 expression and clinicopathological variables of OSCC patients (*n* = 50).

Characteristics	Radiation Therapy Response		*p*-Value
	Non-Recurrence (20)	Radio-Resistance (30)	
Age			0.5637
• <60 yo	11 (55%)	14 (46.6%)	
• ≥60 yo	9 (45%)	16 (53.3%)	
Sex			0.588
• Male	16 (80%)	22 (73.3%)	
• Female	4 (20%)	8 (26.6%)	
Disease Site			0.699
• Oral Cavity	15 (75%)	21 (70%)	
• Oropharynx	5 (25%)	9 (45%)	
Stage			0.0038
• Non-Metastatic	15 (75%)	10 (33.3%)	
• Metastatic Disease	5 (25%)	20 (66.6%)	
Histological Grade			0.00078
• Poor	5 (25%)	22 (73.3%)	
• Well/Moderate	15 (75%)	8 (26.6%)	
DYRK3 expression			0.00016
• Low	12 (60%)	3 (10%)	
• High	8 (40%)	27 (90%)	
PAICS expression			0.368
• Low	7 (35%)	7 (23.3%)	
• High	13 (65%)	23 (76.6%)	
PPAT expression			0.001
• Low	12 (60%)	5 (16.6%)	
• High	8 (40%)	25 (83.3%)	

## Data Availability

Experimental procedures, characterization of new compounds, and all other data supporting the findings are available in the Appendix A. All authors provide consent for all the aspects of this work to ensure consistency and accuracy, and they have endorsed a final submitted version of the manuscript. The datasets that are used and analyzed in this investigation will be provided by the corresponding author upon reasonable request.

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
