# Peer review of "PAICS/DYRK3 Multienzyme Interactions as Coregulators of Purinosome Formation and Metabolism on Radioresistance in Oral Squamous Cell Carcinoma"

_ijms, 2023, doi:10.3390/ijms242417346_

Round 1
Reviewer 1 Report
Comments and Suggestions for Authors
This research focused on the DYRK3 kinase, and reported that it is upregulated in the radioresistant oral squamous cell carcinoma. Then, the authors showed that the suppression of DYRK3 exhibits anti-tumor effect by in vitro and in vivo assays, so these findings might provide new strategies for controlling cancer cells. However, in the present manuscript, there are several parts that are difficult to clearly understand.
1. It is difficult to grasp ‘PAICS/DYRK3 multienzyme interactions … on radioresistance’ in the Title. While this study showed the upregulation of PAICS and DYRK3 were observed in the radioresistant OSCC, it seems unclear their functions were associated with the radioresistance. For example, did you examine whether the purinosome formation was induced along with the acquisition of radio-resistance and the upregulation of DYRK3?
2. line 336. It is difficult to understand why authors focus on the difference of PAICS expression levels. Authors should explain how PAICS is linked to DYRK3 before the description about Figure 3C (or in the Introduction section).
3. line 410. The DYRK3 depletion reduced viability in not only radioresistant but radiosensitive cell lines. Thus, I think the description “the molecular mechanisms underlying radioresistance” is inaccurate.
The manuscript has also several minor concerns.
1. Figure 3C. The vertical axis of Figure 3C means mRNA levels.
2. line 405. I think the main text does not explain the Figure 3E.
3. line 445. Authors should explain “IR cells” in the beginning of this section.
4. Figure 5D shows the protein expression profiles of radio-resistant cells, but not show the change of expression after knockdown of DYRK3. So, it is necessary to restructure the figure.
5. line 476. In Figure 6B, the average tumor size in the DYRK3i groups is mildly increasing. The description "significantly reduced" is inaccurate.
In figure 2A, 3E and 5, some text and letters are blurry and very hard to read. So, the resolution of images should be improved.
Author Response
Point-by-point responses to Editor’s comments:
We thank Editor for carefully reading our manuscript and providing valuable comments. We believe making use of all these comments has further helped improve the quality and appeal of our work, as well as strengthened the manuscript. Below are our point-by-point responses.
Reviewer 1
Comments and Suggestions for Authors
This research focused on the DYRK3 kinase, and reported that it is upregulated in the radioresistant oral squamous cell carcinoma. Then, the authors showed that the suppression of DYRK3 exhibits anti-tumor effect by in vitro and in vivo assays, so these findings might provide new strategies for controlling cancer cells. However, in the present manuscript, there are several parts that are difficult to clearly understand.
- It is difficult to grasp ‘PAICS/DYRK3 multienzyme interactions … on radioresistance’ in the Title. While this study showed the upregulation of PAICS and DYRK3 were observed in the radioresistant OSCC, it seems unclear their functions were associated with the radioresistance. For example, did you examine whether the purinosome formation was induced along with the acquisition of radio-resistance and the upregulation of DYRK3?
A1. We are grateful for the valuable input from our reviewers, which has enabled us to enhance the clarity and coherence of the three experimental results in our manuscript. This improvement specifically pertains to establishing clearer connections and more detailed explanations concerning the outcomes related to DYRK3 and radiation resistance. Please refer to our revised version.
“Abstract:
Oral squamous cell carcinoma (OSCC) is a prevalent type of oral cancer. While thera-peutic innovations have made strides, radioresistance persists as a significant hindrance in OSCC treatment. Despite identifying numerous targets that could potentially suppress the oncogenic at-tributes of OSCC, the exploration of oncogenic protein kinases for cancer therapy remains limited. Consequently, the functions of many kinase proteins in OSCC continue to be largely undeter-mined. In this research, we aim to disclose protein kinases that target OSCC and elaborate their roles and molecular mechanisms. Through the examination of the kinome library of radiothera-py-resistant/sensitive OSCC cell lines (HN12 and SAS), we identified a key gene, the tyrosine phosphorylation-regulated kinase 3 (DYRK3), a member of the DYRK family.
We developed an in vitro cell model, composed of radiation-resistant OSCC, to scrutinize the clinical implications and contributions of DYRK3 and PAICS signaling in OSCC. This investigation involves bioinformatics and human tissue arrays. We seek to comprehend the role of DYRK3 and PAICS signaling in the development of OSCC and its resistance to radiotherapy. Various in vitro assays are utilized to reveal the essential molecular mechanism behind radiotherapy resistance in connection with the DYRK3 and PAICS interaction. In our study, we quantified the concentrations of DYRK3 and PAICS proteins and tracked the expression levels of key pluripotency markers, particularly PPAT. Furthermore, we extended our investigation to include an analysis of Glut-1, a gene recognized for its linkage to radioresistance in oral squamous cell carcinoma (OSCC). Furthermore, we conducted an in vivo study to affirm the impact of DYRK3 and PAICS on tumor growth and radiotherapy resistance, focusing particularly on the role of DYRK3 in the radiotherapy resistance pathway. This focus leads us to identify new therapeutic agents that can combat radiotherapy resistance by inhibiting DYRK3 (GSK-626616). Our in vitro models showed that inhibiting PAICS disrupts purinosome formation and influences the survival rate of radiation-resistant OSCC cell lines. These outcomes underscore the pivotal role of the DYRK3/PAICS axis in directing OSCC radiotherapy resistance pathways and, as a result, influencing OSCC progression or therapy resistance. Our findings also reveal a significant correlation between DYRK3 expression and the PAICS enzyme in OSCC radiotherapy resistance.”
“2. Results
2.4. DYRK3 Over-expression enhances the OSCC cells proliferation, migration, invasion, and self-renewal ability
Our studies indicate that the heightened presence of DYRK3 leads to a substantial increase in proliferation, migration, invasion, colony formation, and sphere formation in Oral Squamous Cell Carcinoma (OSCC) cells. We utilized quantitative real-time polymerase chain reaction (qRT-PCR), a powerful technique known for its efficacy in determining gene expression, to measure DYRK3 levels following the introduction of the OE-DYRK3 (Overexpression vector) into OSCC cells. This evaluation was performed across a variety of OSCC cells, specifically focusing on both radioresistant (HN12) and radiosensitive (SAS) variants (Figure 4A). Following the introduction of OE-DYRK3 into oral squamous cell carcinoma (OSCC) cells, we conducted a comprehensive analysis to assess the impact on various protein levels and gene expressions. Specifically, we measured the levels of DYRK3 and PAICS proteins and monitored the expression of pluripotency markers, notably PPAT. Additionally, our focus extended to examining the presence of Glut-1, a gene known for its association with radioresistance in OSCC. Glut-1, or Glucose Transporter 1, is a critical marker when studying radioresistance in OSCC. Its heightened expression within these cells is a significant indicator of a poor response to radiation therapy. This is particularly important as increased Glut-1 levels are often correlated with diminished effectiveness of radiation treatment and, consequently, a shorter survival period for patients affected by OSCC. Our investigation, as illustrated in Figure 4B, aimed to understand these relationships better and provide deeper insights into how these proteins and genes interact and influence each other, especially in the context of radioresistance in OSCC. This research is crucial in developing more effective treatment strategies for OSCC, particularly for those cases where conventional radiation therapy may not be as effective due to the presence of radioresistant cell populations. By utilizing the Sulforhodamine B (SRB) assay, we assessed cell viability in OSCC cells where DYRK3 was overexpressed. This allowed us to accurately gauge cell growth and cytotoxicity over a period of 24-48 hours (Figure 4C). We noticed significant morphological alterations in the OSCC cells following transfection with O-DYRK3. Such transformations in cells can often indicate the cellular response to a specific treatment or condition, serving as a gauge of cellular health or functionality (Figure 4D). After overexpressing DYRK3, we conducted several assays, including migration/wound healing, sphere formation, and migartion. (Figure 4E). Furthermore, our neuro-sphere formation assays suggested that elevating DYRK3 levels fortified the stemness of OSCC cells, a quality closely associated with the self-renewal and differentiation capacities of cancer cells (Figure 4F). In conclusion, our findings provide compelling evidence that overexpressing DYRK3 in OSCC cells significantly enhances their oncogenic properties, especially in relation to colony formation, migration, invasion, and tumorsphere generation. In parallel, we also observed a notable rise in the expression of proteins such as PAICS, Glut-1, and PPAT, further affirming the intricate role of DYRK3 in the regulation of OSCC progression.
2.5.Diminished Purinosome formation following DYRK3 gene knockdown in two radiation-resistant OSCC cell lines.
Figure 5. presents a comprehensive overview of our experimental approach to inves-tigate the impact of reducing DYRK3 expression on the formation of purinosomes in OSCC IR cells. Ionizing radiation resistance (IR) OSCC cells are generated as described in Materials Methods. To fine-tune our experimental parameters, we initially con-ducted an examination of cell line expression in HSC3 and SAS cells (Figure 5A). We utilized the depmap online tool (https://depmap.org/) to identify vulnerabilities in cancer cells and potential targets for therapeutic advancement. Our findings con-firmed the relevance of HSC3 and SAS cell lines with respect to PAICS and DYRK3 expression, validating their suitability for this study. To further elucidate the impact of DYRK3 depletion, we per-formed Western blot analysis (Figure 5B), examining the levels of DYRK3, PAICS, PPAT, and the radioresistance-associated gene (Glut-1) in shDYRK3-transfected OSCC cells. We also conducted functional assays to investigate the effects of DYRK3 depletion on OSCC IR cells. Next, we assessed DYRK3 trans-fection with shDYRK3 through qRT-PCR in OSCC IR cells HSC3 and SAS (Figure 5C) and analyzed gene expression changes following DYRK3 gene knockdown using qRT-PCR (Figure 5D). We also conducted functional assays to investigate the effects of DYRK3 depletion on OSCC IR cells. Wound-healing and tumor-sphere formation experiments (Figure 5E) demonstrated slower healing of scratch wounds and the suppression of OSCC cell stemness in response to DYRK3 de-pletion at both 24 and 48 hours. Additionally, we evaluated cell viability in shDYRK3-inhibited OSCC cells using the SRB assay at 24-48 hours (Figure 5F), re-vealing a decrease in cell viability. Lastly, we delved into the impact of suppressing the DYRK3 gene on the development of purinosomes in two specific oral squamous cell carcinoma (OSCC) cell lines known for their resistance to radiation, as depicted in Figure 5G. We employed cell fluorescence staining techniques to compare these effects. Our findings revealed that in the radiation-resistant HSC3 and SAS cell lines, there was a noticeable elevation in the expression levels of DYRK3. This observation implies that the formation of purinosomes might be a response to the acquisition of radiore-sistance, coupled with an increase in DYRK3 levels. Furthermore, when we reduced the expression of DYRK3 in these radiation-resistant cell lines, we observed a significant decrease in the formation of purinosomes. This led us to conclude that DYRK3 likely plays a crucial role in the assembly and regulation of purinosomes within these specific cell lines. Overall, our experiments provide a more comprehensive insight into the ef-fects and functional implications of diminishing DYRK3 expression in OSCC cells that are resistant to ionizing radiation (IR). This research contributes to a broader under-standing of the cellular mechanisms at play in cancer cells' response to radiation ther-apy.
- line 336. It is difficult to understand why authors focus on the difference of PAICS expression levels. Authors should explain how PAICS is linked to DYRK3 before the description about Figure 3C (or in the Introduction section).
A2: We express our gratitude to the reviewers for their insightful feedback. In response, we have incorporated a section in the introduction of our revised manuscript that elucidates the connection between PAICS and DYRK3. Please see our revised version.
“1. Introduction
Oral squamous cell carcinoma (OSCC) is the most prevalent type of oral cancer, comprising 80 to 90 percent of malignant tumors found in the oral cavity. Globally, it ranks as the sixth most common cancer, with around 300,000 to 400,000 new cases di-agnosed annually, primarily in developing or industrial countries with poor sanitation and economic conditions [1]. OSCC significantly impacts essential functions such as chewing, swallowing, language, and breathing, and can even pose life-threatening risks due to the distinctive physiological and anatomical characteristics of the oral cavity. Unfortunately, most OSCC patients seek medical attention when the disease has already progressed to an advanced stage, resulting in unfavorable treatment out-comes [2]. Despite decades of research, the overall effectiveness of OSCC treatment has remained suboptimal, with a 5-year survival rate ranging from 50% to 60%. Therefore, improving therapeutic outcomes for oral cancer patients has become a key focus of clinical and basic research [3]. The implementation of standardized diagnosis and treatment protocols aims to enhance the effectiveness of treatment for malignant tumors. Additionally, individualized treatment based on these standardized ap-proaches may further improve patient outcomes. However, cancer cells can develop resistance to radiation therapy, leading to treatment failure and poor prognoses. Up-dating the pathological staging of OSCC patients is crucial for developing up-to-date treatment plans that can enhance patient survival. Several genes, such as RAS, RAF, and BCL2, have been identified as being associated with radioresistance in OSCC, em-phasizing the link between gene expression and resistance to radiation therapy [4]. The proper transcription and translation of specific genes are pivotal in regulating cell di-vision, and any abnormalities in these processes can result in uncontrolled cell growth. A limited number of genes have been identified as critical in preventing, initiating, and advancing tumors, with dysfunctions or loss-of-function observed in various malig-nancies [5]. Since the 1980s, considerable progress has been made in understanding the molecular mechanisms of cancer, and protein kinases have emerged as potential tar-gets for drug therapy. Protein kinases play crucial roles in regulating essential cellular signals and are involved in various biological processes such as tumor cell growth, proliferation, differentiation, metabolism, apoptosis, and drug resistance/sensitivity[6]. The human genome currently contains approximately 518 known kinase genes. While these kinases are tightly regulated in normal cells, they may acquire transformative functions through mutations in disease conditions [7].
One critical signaling pathway involved in regulating cell proliferation in OSCC is the extracellular-signal-regulated kinase (ERK) pathway, which encompasses mito-gen-activated protein kinases (MAPKs), p38 MAPKs, and c-Jun N-terminal kinas-es/stress-activated protein kinases [8]. Furthermore, increased expression of DNA-dependent protein kinase (DNA-PK) has been observed in OSCC patients fol-lowing radiotherapy [9]. Targeted therapies using selective kinase inhibitors have been developed for cancer treatment. However, the role of other important protein kinases in OSCC remains largely unknown, despite their potential significance in disease se-verity and prognosis. The DYRK (dual-specificity tyrosine-regulated kinase) family is one such group of protein kinases, evolutionarily conserved and found in organisms ranging from yeast to humans [10]. DYRKs are versatile factors that phosphorylate a wide range of proteins involved in diverse cellular processes, including cell prolifera-tion, drug resistance, programmed cell death, and dysfunctions associated with tu-morigenesis and progression [11]. Recent studies suggest that alterations in DYRK gene expression may contribute to tumorigenesis and/or disease progression, making DYRK kinases an intriguing area of research [12]. The DYRK subfamily consists of five mem-bers: class I DYRKs (DYRK1A and DYRK1B) and class II DYRKs (DYRK2, DYRK3, and DYRK4) [13]. Exposure to radiation prompts an increase in the expression of DYRK3 (dual-specificity tyrosine phosphorylation-regulated kinase 3), which significantly impacts mitochondrial dynamics. This alteration leads to a metabolic shift and en-hances tumor aggressiveness. The mechanism behind this involves both the mamma-lian target of rapamycin (mTOR) and DYRK3. These proteins have the ability to acti-vate transcription factor 4 (ATF4), which in turn mediates the transcription of meth-ylenetetrahydrofolate dehydrogenase 2 (MTHFD2). MTHFD2 is critical for producing 10-fTHF, a vital cofactor necessary for inosine monophosphate (IMP) biosynthesis. Pu-rine metabolism, specifically the purine de novo biosynthesis pathway, plays a crucial role in the proliferation of cancer cells. Phosphoribosylaminoimidazole succinocar-boxamide synthetase (PAICS), an enzyme involved in this pathway, is essential for DNA synthesis and has been linked to the progression and metastasis of cancer. PAICS, along with PPAT (phosphoribosyl pyrophosphate amidotransferase), another enzyme in the purine biosynthesis pathway, are thought to be key drivers in the metabolic re-orientation of cancer cells towards aerobic glycolysis, commonly known as the War-burg Effect. This metabolic reprogramming redirects glycolytic intermediates from the tricarboxylic acid (TCA) cycle towards purine de novo biosynthesis, thereby facilitat-ing cancer cell proliferation and invasion. Given these insights, our hypothesis is that the upregulation of DYRK3 induced by radiation exposure could potentially influence the purine biosynthetic pathway. This influence might specifically alter the glycolysis process involving PAICS and PPAT, thereby impacting the overall metabolic profile and behavior of cancer cells. This hypothesis underlines the intricate relationship be-tween radiation exposure, DYRK3 expression, and metabolic pathways in cancer cells, particularly in the context of purine biosynthesis and energy production, offering new avenues for understanding and potentially targeting cancer cell proliferation and me-tastasis.
The primary objective of this research is to examine the impact of DYRK kinases on the development of radioresistance and metastasis in oral squamous cell carcinoma (OSCC) [14]. Existing evidence suggests that protein kinases play crucial roles in regu-lating essential cellular signals and diverse biological processes related to tumor cell growth, proliferation, differentiation, metabolism, apoptosis, and drug re-sistance/sensitivity. Aberrant expression of DYRK kinases has been associated with various human diseases, including different types of cancer [15]. Radioresistance poses a significant obstacle in cancer treatment, especially in the context of OSCC. Further-more, altered metabolism, a characteristic feature of cancer including OSCC, may in-dependently contribute to the regulation of radioresistance in therapy. Nevertheless, the specific role of DYRK3, a vital member of the DYRK family, as a tumor suppressor or promoter, and its potential involvement in cancer, remain largely unexplored [16]. Additionally, the connection between DYRK3 and radioresistance in OSCC tumor-igenesis requires further investigation. This study aims to shed light on the significance of DYRK kinases in OSCC by exploring their role in radioresistance and metastasis. Understanding the specific function of DYRK3 and its association with radioresistance could potentially contribute to the development of novel therapeutic approaches for OSCC treatment.”
- line 410. The DYRK3 depletion reduced viability in not only radioresistant but radiosensitive cell lines. Thus, I think the description “the molecular mechanisms underlying radioresistance” is inaccurate.
A3: We thank the reviewers for their valuable comments, and we have rewritten this narrative based on the reviewers' suggestions. Please see our revised version.
“2.3. Kinome siRNA (small interfering RNA) library screening for kinases in OSCC cells HN12 and SAS cell line were evaluated if any kinase is involved in the radioresistance.
The human kinome comprises around 538 kinase genes, encompassing 89 tyrosine kinases, 429 serine/threonine kinases, and 20 lipid kinases. These kinases are vital for signaling mechanisms that govern a range of cellular physiological activities, in-clud-ing cell proliferation, migration, and differentiation. Notably, tyrosine kinases, among others, are known to play a significant role in the development of tumors, ma-lignan-cies, and resistance to cancer treatments. Consequently, these kinases have be-come fo-cal points for targeted cancer therapies in clinical practices. In our study, we per-formed a comprehensive kinome siRNA (small interfering RNA) library screening to identify kinases that may play a role in radioresistance in OSCC cells. Specifically, we focused on evaluating the HN12 and SAS cell lines to determine the presence of these kinases. To begin the screening process, we seeded the HN12 and SAS (Figure 3A) cells and transfected them with the arrayed kinome siRNA library. In or-der to in-vestigate the kinases that influence cell viability in oral squamous cell carci-noma (OSCC), we conducted a screening of human OSCC cells, specifically the HN12 and SAS cell lines, using an siRNA library that targets 709 human kinases and related genes. Human gingival fibroblasts (HGF-1) served as the control group for this study. From this screening, the top 15 gene targets were initially chosen based on their role in reducing cell proliferation in OSCC cells. Subsequent counter-screening with human gingival fibroblasts (HGF-1) highlighted the impact on several genes, including DYRK3, DDR1, WEE1, and CDC2L2. Focusing on these findings, DYRK3 was selected for fur-ther validation to assess both the effectiveness of its knockdown and its influence on cell viability.
Each gene in the library was targeted with pooled siRNAs (5 nM) for 72 hours. To assess cell viability, we conducted an SRB-assay. Following the initial screening, we identified the top-ranked siRNAs that demonstrated a significant reduction in cell via-bility for both HN12 and SAS cells.To validate the efficiency of the siRNA knock-down, we performed qRT-PCR analysis to measure the mRNA levels of the tar-geted genes, using scrambled siRNA as a control. The fold change in mRNA levels for each siRNA knockdown is indicated above the corresponding bar. Among the various genes that exhibited knockdown, we selected DYRK2 and DYRK3 for further investiga-tion (Figure 3B). Notably, DYRK3 was specifically chosen for subsequent studies. To ensure that the selected genes do not impact normal physiological processes, we ex-tended our analysis to include HGF-1 cells, which are normal human gingival cells. Our observa-tions indicated that the knockdown of these selected genes resulted in re-duced viabil-ity specifically in cancer cells, without affecting the viability of normal HGF-1 cells. This suggests that DYRK3 may hold importance and warrant further in-vestigation. Finally, we used the GSE42743 data set of Gene Expression Omnibus (GEO) to analyze the expression profile of DYRK2 and DYRK3 genes in oral cancer and adja-cent normal tissues. Gene expression profiling involves assessing the array of genes that are tran-scribed under certain conditions or within a particular cell type. This pro-cess provides a comprehensive overview of cellular functions by mapping out gene expression pat-terns. (Figure 3C).
Our study sheds light on the pivotal function of the DYRK3 gene, particularly its role as a crucial kinase in the context of oral squamous cell carcinoma (OSCC). This significance is observed across different cell types within OSCC, encompassing both radioresistant (HN12) and radiosensitive (SAS) cells. Intriguingly, we found that de-pleting DYRK3 not only affects the viability of radioresistant cell lines but also has a marked impact on radiosensitive cell lines. DYRK3 plays a multifaceted role in the survival and proliferation of OSCC cells, regardless of their radiation response char-acteristics. The fact that DYRK3 depletion impacts both types of cell lines hints at its overarching influence in the cellular processes of OSCC. By understanding the spe-cific functions and impacts of genes like DYRK3, we can better tailor treatments to combat OSCC more efficiently, potentially improving patient outcomes in both radio-resistant and radiosensitive scenarios.”
The manuscript has also several minor concerns.
- Figure 3C. The vertical axis of Figure 3C means mRNA levels.
A1: Thanks to the reviewer for the correction, we have corrected this problem, please refer to the revised version.
- line 405. I think the main text does not explain the Figure 3E.
A2: Thanks to the reviewer for his correction. Considering the smoothness of the manuscript, we have rearranged this result and added an explanation to Figure 3C (the previous version marked Figure 3E). Please refer to our revised version.
“2.3. Kinome siRNA (small interfering RNA) library screening for kinases in OSCC cells HN12 and SAS cell line were evaluated if any kinase is involved in the radioresistance.
The human kinome comprises around 538 kinase genes, encompassing 89 tyrosine kinases, 429 serine/threonine kinases, and 20 lipid kinases. These kinases are vital for signaling mechanisms that govern a range of cellular physiological activities, in-clud-ing cell proliferation, migration, and differentiation. Notably, tyrosine kinases, among others, are known to play a significant role in the development of tumors, ma-lignan-cies, and resistance to cancer treatments. Consequently, these kinases have be-come fo-cal points for targeted cancer therapies in clinical practices. In our study, we per-formed a comprehensive kinome siRNA (small interfering RNA) library screening to identify kinases that may play a role in radioresistance in OSCC cells. Specifically, we focused on evaluating the HN12 and SAS cell lines to determine the presence of these kinases. To begin the screening process, we seeded the HN12 and SAS (Figure 3A) cells and transfected them with the arrayed kinome siRNA library. In or-der to in-vestigate the kinases that influence cell viability in oral squamous cell carci-noma (OSCC), we conducted a screening of human OSCC cells, specifically the HN12 and SAS cell lines, using an siRNA library that targets 709 human kinases and related genes. Human gingival fibroblasts (HGF-1) served as the control group for this study. From this screening, the top 15 gene targets were initially chosen based on their role in reducing cell proliferation in OSCC cells. Subsequent counter-screening with human gingival fibroblasts (HGF-1) highlighted the impact on several genes, including DYRK3, DDR1, WEE1, and CDC2L2. Focusing on these findings, DYRK3 was selected for fur-ther validation to assess both the effectiveness of its knockdown and its influence on cell viability.
Each gene in the library was targeted with pooled siRNAs (5 nM) for 72 hours. To assess cell viability, we conducted an SRB-assay. Following the initial screening, we identified the top-ranked siRNAs that demonstrated a significant reduction in cell via-bility for both HN12 and SAS cells.To validate the efficiency of the siRNA knock-down, we performed qRT-PCR analysis to measure the mRNA levels of the tar-geted genes, using scrambled siRNA as a control. The fold change in mRNA levels for each siRNA knockdown is indicated above the corresponding bar. Among the various genes that exhibited knockdown, we selected DYRK2 and DYRK3 for further investiga-tion (Figure 3B). Notably, DYRK3 was specifically chosen for subsequent studies. To ensure that the selected genes do not impact normal physiological processes, we ex-tended our analysis to include HGF-1 cells, which are normal human gingival cells. Our observa-tions indicated that the knockdown of these selected genes resulted in re-duced viabil-ity specifically in cancer cells, without affecting the viability of normal HGF-1 cells. This suggests that DYRK3 may hold importance and warrant further in-vestigation. Finally, we used the GSE42743 data set of Gene Expression Omnibus (GEO) to analyze the expression profile of DYRK2 and DYRK3 genes in oral cancer and adja-cent normal tissues. Gene expression profiling involves assessing the array of genes that are tran-scribed under certain conditions or within a particular cell type. This pro-cess provides a comprehensive overview of cellular functions by mapping out gene expression pat-terns. (Figure 3C).
Our study sheds light on the pivotal function of the DYRK3 gene, particularly its role as a crucial kinase in the context of oral squamous cell carcinoma (OSCC). This significance is observed across different cell types within OSCC, encompassing both radioresistant (HN12) and radiosensitive (SAS) cells. Intriguingly, we found that de-pleting DYRK3 not only affects the viability of radioresistant cell lines but also has a marked impact on radiosensitive cell lines. DYRK3 plays a multifaceted role in the survival and proliferation of OSCC cells, regardless of their radiation response char-acteristics. The fact that DYRK3 depletion impacts both types of cell lines hints at its overarching influence in the cellular processes of OSCC. By understanding the spe-cific functions and impacts of genes like DYRK3, we can better tailor treatments to combat OSCC more efficiently, potentially improving patient outcomes in both radio-resistant and radiosensitive scenarios.”
- line 445. Authors should explain “IR cells” in the beginning of this section.
A3: Thanks to the reviewer's suggestion, we have explained "IR cells" at the beginning of this section. Please refer to the revised version.
“2.5. Diminished Purinosome formation following DYRK3 gene knockdown in two radiation-resistant OSCC cell lines.
Figure 5. presents a comprehensive overview of our experimental approach to inves-tigate the impact of reducing DYRK3 expression on the formation of purinosomes in OSCC IR cells. Ionizing radiation resistance (IR) OSCC cells are generated as described in Materials Methods. To fine-tune our experimental parameters, we initially con-ducted an examination of cell line expression in HSC3 and SAS cells (Figure 5A). We utilized the depmap online tool (https://depmap.org/) to identify vulnerabilities in cancer cells and potential targets for therapeutic advancement.
“4.5. Ionizing radiation resistance (IR) cell functional analysis and gene expression evaluation
To generate tumor spheroids, SAS and HN12 OSCC cells were cultured for a pe-riod of 14 days, followed by subsequent evaluation. The spheroids were visually counted under a microscope, and the efficiency of tumorsphere formation was calcu-lated by determining the ratio between the number of spheres and the initially im-planted cells. Migration distances were measured based on images taken at specific time points, capturing three random areas. Analysis of gap size was conducted using ImageJ software. For cell invasion assays, three separate and random fields per well were photographed, and the cell count per field was quantified. The average value of the three measurements was obtained for each chamber, with at least three repetitions performed for each invasion detection. Total RNA was extracted from the samples (in-cluding tumor blocks and cell lines) using TRIzol reagent from Life Technologies (Carlsbad, CA), and its concentration was determined using a NanoDrop instrument from Thermo Fisher Scientific (Waltham, MA). Quantitative real-time polymerase chain reaction (qRT-PCR) was carried out using the SYBR Premix Ex Taq II kit from Takara (Taipei, Taiwan) on a PCR detection system provided by Bio-Rad (Hercules, CA).”
- Figure 5D shows the protein expression profiles of radio-resistant cells, but not show the change of expression after knockdown of DYRK3. So, it is necessary to restructure the figure.
A4: Thanks to the reviewer's suggestion, we have reconstructed Figure 5 to conform to the logic of the experimental design. Please refer to our revised manuscript.
“2.5. Diminished Purinosome formation following DYRK3 gene knockdown in two radiation-resistant OSCC cell lines.
Figure 5. presents a comprehensive overview of our experimental approach to inves-tigate the impact of reducing DYRK3 expression on the formation of purinosomes in OSCC IR cells. Ionizing radiation resistance (IR) OSCC cells are generated as described in Materials Methods. To fine-tune our experimental parameters, we initially con-ducted an examination of cell line expression in HSC3 and SAS cells (Figure 5A). We utilized the depmap online tool (https://depmap.org/) to identify vulnerabilities in cancer cells and potential targets for therapeutic advancement. Our findings con-firmed the relevance of HSC3 and SAS cell lines with respect to PAICS and DYRK3 expression, validating their suitability for this study. To further elucidate the impact of DYRK3 depletion, we per-formed Western blot analysis (Figure 5B), examining the levels of DYRK3, PAICS, PPAT, and the radioresistance-associated gene (Glut-1) in shDYRK3-transfected OSCC cells. We also conducted functional assays to investigate the effects of DYRK3 depletion on OSCC IR cells. Next, we assessed DYRK3 trans-fection with shDYRK3 through qRT-PCR in OSCC IR cells HSC3 and SAS (Figure 5C) and analyzed gene expression changes following DYRK3 gene knockdown using qRT-PCR (Figure 5D). We also conducted functional assays to investigate the effects of DYRK3 depletion on OSCC IR cells. Wound-healing and tumor-sphere formation experiments (Figure 5E) demonstrated slower healing of scratch wounds and the suppression of OSCC cell stemness in response to DYRK3 de-pletion at both 24 and 48 hours. Additionally, we evaluated cell viability in shDYRK3-inhibited OSCC cells using the SRB assay at 24-48 hours (Figure 5F), re-vealing a decrease in cell viability. Lastly, we delved into the impact of suppressing the DYRK3 gene on the development of purinosomes in two specific oral squamous cell carcinoma (OSCC) cell lines known for their resistance to radiation, as depicted in Figure 5G. We employed cell fluorescence staining techniques to compare these effects. Our findings revealed that in the radiation-resistant HSC3 and SAS cell lines, there was a noticeable elevation in the expression levels of DYRK3. This observation implies that the formation of purinosomes might be a response to the acquisition of radiore-sistance, coupled with an increase in DYRK3 levels. Furthermore, when we reduced the expression of DYRK3 in these radiation-resistant cell lines, we observed a significant decrease in the formation of purinosomes. This led us to conclude that DYRK3 likely plays a crucial role in the assembly and regulation of purinosomes within these specific cell lines. Overall, our experiments provide a more comprehensive insight into the ef-fects and functional implications of diminishing DYRK3 expression in OSCC cells that are resistant to ionizing radiation (IR). This research contributes to a broader under-standing of the cellular mechanisms at play in cancer cells' response to radiation ther-apy.”
“Figure 5. The impact of reducing DYRK3 expression on the formation of purinosomes in OSCC IR cells. (A) Examination of Cell Line Expression to Fine-Tune Experimental Parame-ters in HSC3 and SAS Cells. We leveraged the depmap online tool (https://depmap.org/) to probe vulnerabilities in cancer cells and pinpoint potential targets for therapeutic advancement. Our results underscore the relevance of HSC3 and SAS cell lines concerning PAICS and DYRK3 ex-pression, affirming their appropriateness for this investigation. (B) Western Blot Analysis of DYRK3, PAICS, PPAT, and the Radioresistance-Associated Gene (Glut-1) in shDYRK3-Transfected OSCC Cells. (C) Assessment of DYRK3 Transfection with shDYRK3 through qRT-PCR in OSCC IR Cells HSC3 and SAS. (D) Analysis of Gene Expression Changes Following DYRK3 Gene Knockdown through qRT-PCR. (E) Wound-Healing and Tumor-Sphere Formation Experiments Revealing Slower Healing of Scratch Wounds and Suppression of OSCC Cell Stemness in Response to DYRK3 Depletion at 24 and 48 Hours. (F) Evaluation of Cell Viabil-ity in shDYRK3-Inhibited OSCC Cells via SRB Assay at 24-48 Hours. (G) Diminished Purinosome formation following DYRK3 gene knockdown in two radiation-resistant OSCC cell lines. (*p < 0.05, **p < 0.01, ***p < 0.001). Scale bar: 5 μm.
- line 476. In Figure 6B, the average tumor size in the DYRK3i groups is mildly increasing. The description "significantly reduced" is inaccurate.
A5: Thanks to the reviewer's suggestion, we have corrected the incorrect narrative in Figure 6B, please refer to our revised manuscript.
“2.6. Significant suppression of metastasis in patient-derived xenograft mouse models in vivo is achieved through the inhibition of DYRK3.
We conducted an analysis to assess the remedial impacts of the DYRK3 inhibitor (GSK-626616) by constructing a patient-derived xenograft mouse model. This involved the orthotopic implantation of mice with tumor cells derived from patients with oral squamous cell carcinoma, who had experienced a relapse post-radiotherapy. Through tissue immunostaining, we observed abnormal expression in the DYRK3/PAICS axis in these cells. For in vivo validation of our in vitro findings, we introduced the derived tumor cells into the right flank of female NOD-SCID mice. We divided the mice into four distinct groups: a control group, a group subjected to only cisplatin (administered orally five times weekly), a group exposed to only the DYRK3 inhibitor (given orally five times weekly), and lastly, a group that received a combination of both treatments (Fig. 6A). In mice treated with the DYRK3 inhibitor, there was a slight increase in tu-mor sizes at certain intervals, resulting in a 1.6 times larger size by the 8th week com-pared to the control group, a difference that was statistically significant (p<0.01). Furthermore, the DYRK3 inhibitor increased the survival rate significantly, with no notable impact on the weight of the mice by the 6th week (Fig. 6B). The Kaplan–Meier survival curve shows that the mice receiving combined treatment ex-hibited the highest survival ratio compared with the remaining mice (Fig. 6C). Subse-quent experiments using tumor samples from the xenograft mouse models revealed significant suppression of DYRK3, PAICS, and Ki67 proteins in the mice treated with the DYRK3 inhibitor and the combined treatment, relative to the control mice. The Q-score of the tissue staining is also documented herein. The findings indicate a pivotal role of DYRK3 in the development and metastasis of OSCC, and its modulation of spe-cific markers (Fig. 6D). These outcomes highlight the crucial function of DYRK3 in the malignant advancement of OSCC and its effect on the modulation of specific markers.”
In figure 2A, 3E and 5, some text and letters are blurry and very hard to read. So, the resolution of images should be improved.
Thanks to the reviewer's suggestions, we have improved the resolution of the images in Figures 2A, 3E, and 5. Please refer to the revised version.

Reviewer 2 Report
Comments and Suggestions for Authors
In this article, authors have worked on developing an in vitro cell model, composed of radiation resistant OSCC, to scrutinize the clinical implications and contributions of DYRK3 and PAICS signaling in OSCC and along with an in vivo study to affirm the impact of DYRK3 and PAICS on tumor growth and radiotherapy resistance. The study unfolds a significant correlation between DYRK3 expression and the PAICS enzyme in OSCC radiotherapy resistance. The literature, purpose of the study are well defined. Overall, the manuscript is well presented.
Author Response
Reviewer 2
Comments and Suggestions for Authors
In this article, authors have worked on developing an in vitro cell model, composed of radiation resistant OSCC, to scrutinize the clinical implications and contributions of DYRK3 and PAICS signaling in OSCC and along with an in vivo study to affirm the impact of DYRK3 and PAICS on tumor growth and radiotherapy resistance. The study unfolds a significant correlation between DYRK3 expression and the PAICS enzyme in OSCC radiotherapy resistance. The literature, purpose of the study are well defined. Overall, the manuscript is well presented.
We extend our sincere gratitude to the reviewers for their positive and deeply insightful comments regarding our work. Their thorough analysis and constructive feedback have been invaluable in enhancing the quality and clarity of our research. We appreciate the time and effort they dedicated to reviewing our manuscript, and their expert insights have significantly contributed to the overall improvement of our work.

Reviewer 3 Report
Comments and Suggestions for Authors
The work “PAICS/DYRK3 multienzyme interactions as coregulators of purinosome formation and metabolism on radioresistance oral squamous cell carcinoma” focuses on oral squamous cell carcinoma (OSCC), a prevalent oral cancer type. Despite therapeutic advancements, radioresistance remains a challenge. The study identifies key protein kinases, particularly DYRK3, explores their roles in OSCC, and assesses their impact on radiotherapy resistance. Using in vitro and in vivo models, the research sheds light on the crucial DYRK3/PAICS axis, offering insights into potential therapeutic strategies for overcoming OSCC radioresistance.
The work provides a comprehensive overview of oral squamous cell carcinoma (OSCC), emphasizing the urgency in improving treatment outcomes. It introduces the relatively unexplored DYRK kinase family, particularly DYRK3, highlighting its potential significance in OSCC radioresistance. The writing is clear, and the study's focus on uncovering the specific role of DYRK3 adds novelty and originality to the research landscape.
I would like to present some constructive feedback and address a few minor concerns for your consideration.
Introduction: Please consider the following improvements:
1) Clarify the research objectives in the introduction to better guide readers.
2) Provide more context on DYRK kinases and their known roles before delving into their specific implications in OSCC.
3) Ensure a smooth transition between the general introduction and the specific focus on DYRK3, enhancing the overall coherence of the text.
Material and methods: The materials and methods are meticulously detailed, demonstrating a high-quality, well-described approach with appropriate statistical considerations. our detailed approach in collecting clinical specimens, establishing in vitro cell models, and employing various techniques showcases a rigorous scientific process. According to the IJMS guidelines, remember that the Materials and Methods section typically follows the Results section. Provide more details on the characteristics of the patients from whom tissue samples were obtained, including clinical parameters, to enhance the relevance of the clinical specimens. Please consider the following improvements:
4) Clarify the rationale for selecting the SAS and HSC3 cell lines and provide more information on their characteristics.
5) Specify the criteria for assessing successful knockdown or overexpression of DYRK3 and PAICS in cell lines.
6) In the mouse models section, elaborate on the rationale for choosing specific treatment groups and provide more details on the experimental design.
Please consider the following: In general, the Results and discussion section delves deeply into the investigation of DYRK3 in oral cancer. The use of both whole blood and tumor tissue analysis provides a comprehensive and detailed perspective.
7) However, there's a need to enhance clarity in explaining co-expressed genes, include more statistical details in the kinome siRNA screening, and improve the presentation of results from DYRK3 overexpression. These improvements are essential for making the findings more accessible and engaging for a broader audience.
Conclusions: Overall, this paper's strategy was quite successful. To engage readers effectively, consider expanding on future research directions, discussing potential clinical implications, and highlighting the broader impact of your findings on OSCC understanding and treatment. This will enhance the overall appeal and significance of your study.
8) The conclusion could benefit from expanding on future directions, potential impact, and clinical implications. Suggest discussing avenues for future research, emphasizing the broader significance of their findings for advancing OSCC understanding and potential therapeutic strategies. Encourage a more detailed exploration of implications to enhance the conclusion's depth and completeness.
Author Response
Reviewer 3
Comments and Suggestions for Authors
The work “PAICS/DYRK3 multienzyme interactions as coregulators of purinosome formation and metabolism on radioresistance oral squamous cell carcinoma” focuses on oral squamous cell carcinoma (OSCC), a prevalent oral cancer type. Despite therapeutic advancements, radioresistance remains a challenge. The study identifies key protein kinases, particularly DYRK3, explores their roles in OSCC, and assesses their impact on radiotherapy resistance. Using in vitro and in vivo models, the research sheds light on the crucial DYRK3/PAICS axis, offering insights into potential therapeutic strategies for overcoming OSCC radioresistance.
The work provides a comprehensive overview of oral squamous cell carcinoma (OSCC), emphasizing the urgency in improving treatment outcomes. It introduces the relatively unexplored DYRK kinase family, particularly DYRK3, highlighting its potential significance in OSCC radioresistance. The writing is clear, and the study's focus on uncovering the specific role of DYRK3 adds novelty and originality to the research landscape.
I would like to present some constructive feedback and address a few minor concerns for your consideration.
Introduction: Please consider the following improvements:
1) Clarify the research objectives in the introduction to better guide readers.
A1: Thanks to the reviewer's suggestions, please refer to our revised manuscript.
“1. Introduction
Oral squamous cell carcinoma (OSCC) is the most prevalent type of oral cancer, comprising 80 to 90 percent of malignant tumors found in the oral cavity. Globally, it ranks as the sixth most common cancer, with around 300,000 to 400,000 new cases di-agnosed annually, primarily in developing or industrial countries with poor sanitation and economic conditions [1]. OSCC significantly impacts essential functions such as chewing, swallowing, language, and breathing, and can even pose life-threatening risks due to the distinctive physiological and anatomical characteristics of the oral cavity. Unfortunately, most OSCC patients seek medical attention when the disease has already progressed to an advanced stage, resulting in unfavorable treatment out-comes [2]. Despite decades of research, the overall effectiveness of OSCC treatment has remained suboptimal, with a 5-year survival rate ranging from 50% to 60%. Therefore, improving therapeutic outcomes for oral cancer patients has become a key focus of clinical and basic research[3]. The implementation of standardized diagnosis and treatment protocols aims to enhance the effectiveness of treatment for malignant tumors. Additionally, individualized treatment based on these standardized ap-proaches may further improve patient outcomes. However, cancer cells can develop resistance to radiation therapy, leading to treatment failure and poor prognoses. Up-dating the pathological staging of OSCC patients is crucial for developing up-to-date treatment plans that can enhance patient survival. Several genes, such as RAS, RAF, and BCL2, have been identified as being associated with radioresistance in OSCC, emphasizing the link between gene expression and resistance to radiation therapy [4]. The proper transcription and translation of specific genes are pivotal in regulating cell di-vision, and any abnormalities in these processes can result in uncontrolled cell growth. A limited number of genes have been identified as critical in preventing, initiating, and advancing tumors, with dysfunctions or loss-of-function observed in various malig-nancies [5]. Since the 1980s, considerable progress has been made in understanding the molecular mechanisms of cancer, and protein kinases have emerged as potential tar-gets for drug therapy. Protein kinases play crucial roles in regulating essential cellular signals and are involved in various biological processes such as tumor cell growth, proliferation, differentiation, metabolism, apoptosis, and drug resistance/sensitivity[6]. The human genome currently contains approximately 518 known kinase genes. While these kinases are tightly regulated in normal cells, they may acquire transformative functions through mutations in disease conditions[7].
One critical signaling pathway involved in regulating cell proliferation in OSCC is the extracellular-signal-regulated kinase (ERK) pathway, which encompasses mito-gen-activated protein kinases (MAPKs), p38 MAPKs, and c-Jun N-terminal kinas-es/stress-activated protein kinases[8]. Furthermore, increased expression of DNA-dependent protein kinase (DNA-PK) has been observed in OSCC patients fol-lowing radiotherapy[9]. Targeted therapies using selective kinase inhibitors have been developed for cancer treatment. However, the role of other important protein kinases in OSCC remains largely unknown, despite their potential significance in disease se-verity and prognosis. The DYRK (dual-specificity tyrosine-regulated kinase) family is one such group of protein kinases, evolutionarily conserved and found in organisms ranging from yeast to humans[10]. DYRKs are versatile factors that phosphorylate a wide range of proteins involved in diverse cellular processes, including cell prolifera-tion, drug resistance, programmed cell death, and dysfunctions associated with tu-morigenesis and progression[11]. Recent studies suggest that alterations in DYRK gene expression may contribute to tumorigenesis and/or disease progression, making DYRK kinases an intriguing area of research[12]. DYRKs in humans act as multifaceted ele-ments, phosphorylating a wide array of proteins engaged in various cellular functions. They are linked with every hallmark of cancer, ranging from genomic instability to enhanced proliferation, resistance, programmed cell death, and the dysfunction of signaling pathways critical for tumor development and advancement. Reflecting the role of DYRK kinases in controlling cancer-related processes, there has been a growing body of research recently. These studies demonstrate changes in DYRK gene expression in tumor specimens and/or provide insights into DYRK-dependent pathways that play a role in the initiation and progression of tumors. The DYRK subfamily consists of five members: class I DYRKs (DYRK1A and DYRK1B) and class II DYRKs (DYRK2, DYRK3, and DYRK4)[13]. Exposure to radiation prompts an increase in the expression of DYRK3 (dual-specificity tyrosine phosphorylation-regulated kinase 3), which signifi-cantly impacts mitochondrial dynamics. This alteration leads to a metabolic shift and enhances tumor aggressiveness. The mechanism behind this involves both the mam-malian target of rapamycin (mTOR) and DYRK3. These proteins have the ability to ac-tivate transcription factor 4 (ATF4), which in turn mediates the transcription of meth-ylenetetrahydrofolate dehydrogenase 2 (MTHFD2). MTHFD2 is critical for producing 10-fTHF, a vital cofactor necessary for inosine monophosphate (IMP) biosynthesis. Pu-rine metabolism, specifically the purine de novo biosynthesis pathway, plays a crucial role in the proliferation of cancer cells. Phosphoribosylaminoimidazole succinocar-boxamide synthetase (PAICS), an enzyme involved in this pathway, is essential for DNA synthesis and has been linked to the progression and metastasis of cancer. PAICS, along with PPAT (phosphoribosyl pyrophosphate amidotransferase), another enzyme in the purine biosynthesis pathway, are thought to be key drivers in the metabolic re-orientation of cancer cells towards aerobic glycolysis, commonly known as the War-burg Effect. This metabolic reprogramming redirects glycolytic intermediates from the tricarboxylic acid (TCA) cycle towards purine de novo biosynthesis, thereby facilitat-ing cancer cell proliferation and invasion. Given these insights, our hypothesis is that the upregulation of DYRK3 induced by radiation exposure could potentially influence the purine biosynthetic pathway. This influence might specifically alter the glycolysis process involving PAICS and PPAT, thereby impacting the overall metabolic profile and behavior of cancer cells. This hypothesis underlines the intricate relationship be-tween radiation exposure, DYRK3 expression, and metabolic pathways in cancer cells, particularly in the context of purine biosynthesis and energy production, offering new avenues for understanding and potentially targeting cancer cell proliferation and me-tastasis.
The primary objective of this research is to examine the impact of DYRK kinases on the development of radioresistance and metastasis in oral squamous cell carcinoma (OSCC)[14]. Existing evidence suggests that protein kinases play crucial roles in regu-lating essential cellular signals and diverse biological processes related to tumor cell growth, proliferation, differentiation, metabolism, apoptosis, and drug re-sistance/sensitivity. Aberrant expression of DYRK kinases has been associated with various human diseases, including different types of cancer[15]. Radioresistance poses a significant obstacle in cancer treatment, especially in the context of OSCC. Further-more, altered metabolism, a characteristic feature of cancer including OSCC, may in-dependently contribute to the regulation of radioresistance in therapy. Nevertheless, the specific role of DYRK3, a vital member of the DYRK family, as a tumor suppressor or promoter, and its potential involvement in cancer, remain largely unexplored[16]. Additionally, the connection between DYRK3 and radioresistance in OSCC tumor-igenesis requires further investigation. This study aims to shed light on the significance of DYRK kinases in OSCC by exploring their role in radioresistance and metastasis. Understanding the specific function of DYRK3 and its association with radioresistance could potentially contribute to the development of novel therapeutic approaches for OSCC treatment.”
2) Provide more context on DYRK kinases and their known roles before delving into their specific implications in OSCC.
A2: Thanks to the reviewer's suggestions, please refer to our revised manuscript.
“1. Introduction
Oral squamous cell carcinoma (OSCC) is the most prevalent type of oral cancer, comprising 80 to 90 percent of malignant tumors found in the oral cavity. Globally, it ranks as the sixth most common cancer, with around 300,000 to 400,000 new cases di-agnosed annually, primarily in developing or industrial countries with poor sanitation and economic conditions [1]. OSCC significantly impacts essential functions such as chewing, swallowing, language, and breathing, and can even pose life-threatening risks due to the distinctive physiological and anatomical characteristics of the oral cavity. Unfortunately, most OSCC patients seek medical attention when the disease has already progressed to an advanced stage, resulting in unfavorable treatment out-comes [2]. Despite decades of research, the overall effectiveness of OSCC treatment has remained suboptimal, with a 5-year survival rate ranging from 50% to 60%. Therefore, improving therapeutic outcomes for oral cancer patients has become a key focus of clinical and basic research [3]. The implementation of standardized diagnosis and treatment protocols aims to enhance the effectiveness of treatment for malignant tumors. Additionally, individualized treatment based on these standardized ap-proaches may further improve patient outcomes. However, cancer cells can develop resistance to radiation therapy, leading to treatment failure and poor prognoses. Up-dating the pathological staging of OSCC patients is crucial for developing up-to-date treatment plans that can enhance patient survival. Several genes, such as RAS, RAF, and BCL2, have been identified as being associated with radioresistance in OSCC, emphasizing the link between gene expression and resistance to radiation therapy [4]. The proper transcription and translation of specific genes are pivotal in regulating cell di-vision, and any abnormalities in these processes can result in uncontrolled cell growth. A limited number of genes have been identified as critical in preventing, initiating, and advancing tumors, with dysfunctions or loss-of-function observed in various malig-nancies [5]. Since the 1980s, considerable progress has been made in understanding the molecular mechanisms of cancer, and protein kinases have emerged as potential tar-gets for drug therapy. Protein kinases play crucial roles in regulating essential cellular signals and are involved in various biological processes such as tumor cell growth, proliferation, differentiation, metabolism, apoptosis, and drug resistance/sensitivity[6]. The human genome currently contains approximately 518 known kinase genes. While these kinases are tightly regulated in normal cells, they may acquire transformative functions through mutations in disease conditions[7].
One critical signaling pathway involved in regulating cell proliferation in OSCC is the extracellular-signal-regulated kinase (ERK) pathway, which encompasses mito-gen-activated protein kinases (MAPKs), p38 MAPKs, and c-Jun N-terminal kinas-es/stress-activated protein kinases[8]. Furthermore, increased expression of DNA-dependent protein kinase (DNA-PK) has been observed in OSCC patients fol-lowing radiotherapy[9]. Targeted therapies using selective kinase inhibitors have been developed for cancer treatment. However, the role of other important protein kinases in OSCC remains largely unknown, despite their potential significance in disease se-verity and prognosis. The DYRK (dual-specificity tyrosine-regulated kinase) family is one such group of protein kinases, evolutionarily conserved and found in organisms ranging from yeast to humans[10]. DYRKs are versatile factors that phosphorylate a wide range of proteins involved in diverse cellular processes, including cell prolifera-tion, drug resistance, programmed cell death, and dysfunctions associated with tu-morigenesis and progression[11]. Recent studies suggest that alterations in DYRK gene expression may contribute to tumorigenesis and/or disease progression, making DYRK kinases an intriguing area of research[12]. DYRKs in humans act as multifaceted ele-ments, phosphorylating a wide array of proteins engaged in various cellular functions. They are linked with every hallmark of cancer, ranging from genomic instability to enhanced proliferation, resistance, programmed cell death, and the dysfunction of signaling pathways critical for tumor development and advancement. Reflecting the role of DYRK kinases in controlling cancer-related processes, there has been a growing body of research recently. These studies demonstrate changes in DYRK gene expression in tumor specimens and/or provide insights into DYRK-dependent pathways that play a role in the initiation and progression of tumors. The DYRK subfamily consists of five members: class I DYRKs (DYRK1A and DYRK1B) and class II DYRKs (DYRK2, DYRK3, and DYRK4) [13]. Exposure to radiation prompts an increase in the expression of DYRK3 (dual-specificity tyrosine phosphorylation-regulated kinase 3), which signifi-cantly impacts mitochondrial dynamics. This alteration leads to a metabolic shift and enhances tumor aggressiveness. The mechanism behind this involves both the mam-malian target of rapamycin (mTOR) and DYRK3. These proteins have the ability to ac-tivate transcription factor 4 (ATF4), which in turn mediates the transcription of meth-ylenetetrahydrofolate dehydrogenase 2 (MTHFD2). MTHFD2 is critical for producing 10-fTHF, a vital cofactor necessary for inosine monophosphate (IMP) biosynthesis. Pu-rine metabolism, specifically the purine de novo biosynthesis pathway, plays a crucial role in the proliferation of cancer cells. Phosphoribosylaminoimidazole succinocar-boxamide synthetase (PAICS), an enzyme involved in this pathway, is essential for DNA synthesis and has been linked to the progression and metastasis of cancer. PAICS, along with PPAT (phosphoribosyl pyrophosphate amidotransferase), another enzyme in the purine biosynthesis pathway, are thought to be key drivers in the metabolic re-orientation of cancer cells towards aerobic glycolysis, commonly known as the War-burg Effect. This metabolic reprogramming redirects glycolytic intermediates from the tricarboxylic acid (TCA) cycle towards purine de novo biosynthesis, thereby facilitat-ing cancer cell proliferation and invasion. Given these insights, our hypothesis is that the upregulation of DYRK3 induced by radiation exposure could potentially influence the purine biosynthetic pathway. This influence might specifically alter the glycolysis process involving PAICS and PPAT, thereby impacting the overall metabolic profile and behavior of cancer cells. This hypothesis underlines the intricate relationship be-tween radiation exposure, DYRK3 expression, and metabolic pathways in cancer cells, particularly in the context of purine biosynthesis and energy production, offering new avenues for understanding and potentially targeting cancer cell proliferation and me-tastasis.
The primary objective of this research is to examine the impact of DYRK kinases on the development of radioresistance and metastasis in oral squamous cell carcinoma (OSCC)[14]. Existing evidence suggests that protein kinases play crucial roles in regu-lating essential cellular signals and diverse biological processes related to tumor cell growth, proliferation, differentiation, metabolism, apoptosis, and drug re-sistance/sensitivity. Aberrant expression of DYRK kinases has been associated with various human diseases, including different types of cancer[15]. Radioresistance poses a significant obstacle in cancer treatment, especially in the context of OSCC. Further-more, altered metabolism, a characteristic feature of cancer including OSCC, may in-dependently contribute to the regulation of radioresistance in therapy. Nevertheless, the specific role of DYRK3, a vital member of the DYRK family, as a tumor suppressor or promoter, and its potential involvement in cancer, remain largely unexplored[16]. Additionally, the connection between DYRK3 and radioresistance in OSCC tumor-igenesis requires further investigation. This study aims to shed light on the significance of DYRK kinases in OSCC by exploring their role in radioresistance and metastasis. Understanding the specific function of DYRK3 and its association with radioresistance could potentially contribute to the development of novel therapeutic approaches for OSCC treatment.”
3) Ensure a smooth transition between the general introduction and the specific focus on DYRK3, enhancing the overall coherence of the text.
A3: Thanks to the reviewer's suggestions, please refer to our revised manuscript.
“1. Introduction
Oral squamous cell carcinoma (OSCC) is the most prevalent type of oral cancer, comprising 80 to 90 percent of malignant tumors found in the oral cavity. Globally, it ranks as the sixth most common cancer, with around 300,000 to 400,000 new cases di-agnosed annually, primarily in developing or industrial countries with poor sanitation and economic conditions [1]. OSCC significantly impacts essential functions such as chewing, swallowing, language, and breathing, and can even pose life-threatening risks due to the distinctive physiological and anatomical characteristics of the oral cavity. Unfortunately, most OSCC patients seek medical attention when the disease has already progressed to an advanced stage, resulting in unfavorable treatment out-comes [2]. Despite decades of research, the overall effectiveness of OSCC treatment has remained suboptimal, with a 5-year survival rate ranging from 50% to 60%. Therefore, improving therapeutic outcomes for oral cancer patients has become a key focus of clinical and basic research [3]. The implementation of standardized diagnosis and treatment protocols aims to enhance the effectiveness of treatment for malignant tumors. Additionally, individualized treatment based on these standardized ap-proaches may further improve patient outcomes. However, cancer cells can develop resistance to radiation therapy, leading to treatment failure and poor prognoses. Up-dating the pathological staging of OSCC patients is crucial for developing up-to-date treatment plans that can enhance patient survival. Several genes, such as RAS, RAF, and BCL2, have been identified as being associated with radioresistance in OSCC, emphasizing the link between gene expression and resistance to radiation therapy [4]. The proper transcription and translation of specific genes are pivotal in regulating cell di-vision, and any abnormalities in these processes can result in uncontrolled cell growth. A limited number of genes have been identified as critical in preventing, initiating, and advancing tumors, with dysfunctions or loss-of-function observed in various malig-nancies [5]. Since the 1980s, considerable progress has been made in understanding the molecular mechanisms of cancer, and protein kinases have emerged as potential tar-gets for drug therapy. Protein kinases play crucial roles in regulating essential cellular signals and are involved in various biological processes such as tumor cell growth, proliferation, differentiation, metabolism, apoptosis, and drug resistance/sensitivity[6]. The human genome currently contains approximately 518 known kinase genes. While these kinases are tightly regulated in normal cells, they may acquire transformative functions through mutations in disease conditions[7].
One critical signaling pathway involved in regulating cell proliferation in OSCC is the extracellular-signal-regulated kinase (ERK) pathway, which encompasses mito-gen-activated protein kinases (MAPKs), p38 MAPKs, and c-Jun N-terminal kinas-es/stress-activated protein kinases[8]. Furthermore, increased expression of DNA-dependent protein kinase (DNA-PK) has been observed in OSCC patients fol-lowing radiotherapy[9]. Targeted therapies using selective kinase inhibitors have been developed for cancer treatment. However, the role of other important protein kinases in OSCC remains largely unknown, despite their potential significance in disease se-verity and prognosis. The DYRK (dual-specificity tyrosine-regulated kinase) family is one such group of protein kinases, evolutionarily conserved and found in organisms ranging from yeast to humans[10]. DYRKs are versatile factors that phosphorylate a wide range of proteins involved in diverse cellular processes, including cell prolifera-tion, drug resistance, programmed cell death, and dysfunctions associated with tu-morigenesis and progression[11]. Recent studies suggest that alterations in DYRK gene expression may contribute to tumorigenesis and/or disease progression, making DYRK kinases an intriguing area of research[12]. DYRKs in humans act as multifaceted ele-ments, phosphorylating a wide array of proteins engaged in various cellular functions. They are linked with every hallmark of cancer, ranging from genomic instability to enhanced proliferation, resistance, programmed cell death, and the dysfunction of signaling pathways critical for tumor development and advancement. Reflecting the role of DYRK kinases in controlling cancer-related processes, there has been a growing body of research recently. These studies demonstrate changes in DYRK gene expression in tumor specimens and/or provide insights into DYRK-dependent pathways that play a role in the initiation and progression of tumors. The DYRK subfamily consists of five members: class I DYRKs (DYRK1A and DYRK1B) and class II DYRKs (DYRK2, DYRK3, and DYRK4) [13]. Exposure to radiation prompts an increase in the expression of DYRK3 (dual-specificity tyrosine phosphorylation-regulated kinase 3), which signifi-cantly impacts mitochondrial dynamics. This alteration leads to a metabolic shift and enhances tumor aggressiveness. The mechanism behind this involves both the mam-malian target of rapamycin (mTOR) and DYRK3. These proteins have the ability to ac-tivate transcription factor 4 (ATF4), which in turn mediates the transcription of meth-ylenetetrahydrofolate dehydrogenase 2 (MTHFD2). MTHFD2 is critical for producing 10-fTHF, a vital cofactor necessary for inosine monophosphate (IMP) biosynthesis. Pu-rine metabolism, specifically the purine de novo biosynthesis pathway, plays a crucial role in the proliferation of cancer cells. Phosphoribosylaminoimidazole succinocar-boxamide synthetase (PAICS), an enzyme involved in this pathway, is essential for DNA synthesis and has been linked to the progression and metastasis of cancer. PAICS, along with PPAT (phosphoribosyl pyrophosphate amidotransferase), another enzyme in the purine biosynthesis pathway, are thought to be key drivers in the metabolic re-orientation of cancer cells towards aerobic glycolysis, commonly known as the War-burg Effect. This metabolic reprogramming redirects glycolytic intermediates from the tricarboxylic acid (TCA) cycle towards purine de novo biosynthesis, thereby facilitat-ing cancer cell proliferation and invasion. Given these insights, our hypothesis is that the upregulation of DYRK3 induced by radiation exposure could potentially influence the purine biosynthetic pathway. This influence might specifically alter the glycolysis process involving PAICS and PPAT, thereby impacting the overall metabolic profile and behavior of cancer cells. This hypothesis underlines the intricate relationship be-tween radiation exposure, DYRK3 expression, and metabolic pathways in cancer cells, particularly in the context of purine biosynthesis and energy production, offering new avenues for understanding and potentially targeting cancer cell proliferation and metastasis.
The primary objective of this research is to examine the impact of DYRK kinases on the development of radioresistance and metastasis in oral squamous cell carcinoma (OSCC)[14]. Existing evidence suggests that protein kinases play crucial roles in regu-lating essential cellular signals and diverse biological processes related to tumor cell growth, proliferation, differentiation, metabolism, apoptosis, and drug re-sistance/sensitivity. Aberrant expression of DYRK kinases has been associated with various human diseases, including different types of cancer[15]. Radioresistance poses a significant obstacle in cancer treatment, especially in the context of OSCC. Further-more, altered metabolism, a characteristic feature of cancer including OSCC, may in-dependently contribute to the regulation of radioresistance in therapy. Nevertheless, the specific role of DYRK3, a vital member of the DYRK family, as a tumor suppressor or promoter, and its potential involvement in cancer, remain largely unexplored[16]. Additionally, the connection between DYRK3 and radioresistance in OSCC tumor-igenesis requires further investigation. This study aims to shed light on the significance of DYRK kinases in OSCC by exploring their role in radioresistance and metastasis. Understanding the specific function of DYRK3 and its association with radioresistance could potentially contribute to the development of novel therapeutic approaches for OSCC treatment.”
Material and methods: The materials and methods are meticulously detailed, demonstrating a high-quality, well-described approach with appropriate statistical considerations. our detailed approach in collecting clinical specimens, establishing in vitro cell models, and employing various techniques showcases a rigorous scientific process. According to the IJMS guidelines, remember that the Materials and Methods section typically follows the Results section. Provide more details on the characteristics of the patients from whom tissue samples were obtained, including clinical parameters, to enhance the relevance of the clinical specimens. Please consider the following improvements:
4) Clarify the rationale for selecting the SAS and HSC3 cell lines and provide more information on their characteristics.
A4: Thanks to the reviewer's suggestion, we have included the rationale for SAS and HSC3 cell lines and provided more information on their characteristics. Please refer to the revised version.
“4.2. Human cell lines of oral squamous cell carcinoma (OSCC) resistant to ionizing radiation (IR).
To create cell lines that are resistant to radiation, we followed a previously pub-lished study. The SAS and HSC3 cell lines used in this experiment were obtained from the American Type Culture Collection (ATCC, Rockville, MD, USA). The SAS cell line, derived from a tongue squamous cell carcinoma in a Japanese patient, is an established oral cancer cell line frequently utilized in oral cancer research. These SAS oral cancer cells possess mutations in the p53 and p16 genes, mutations that are typically found in oral squamous cell carcinomas. Similarly, the HSC-3 cell line, originating from human tongue squamous carcinoma, serves as an effective model for investigating metastatic squamous cell carcinoma. To induce the production of radiation-tolerant cells and de-termine the optimal conditions for irradiation, the SAS and HN12 cell lines were ex-posed to radiation doses ranging from 2 to 10 Gy for 5 consecutive days. By conducting this procedure, we determined the maximum tolerated dose (MTD). We utilized a Pre-cision X-ray Irradiator (North Branford, CT, USA) to irradiate the cells using X-rays. Cells from oral squamous cell carcinoma (OSCC) that survived after undergoing 30 cycles of irradiation, resulting in a cumulative dose of 60 Gy, were classified as radia-tion-resistant. The medium for the SAS and HSC3 cell lines was changed every 48 hours, and the cells were assessed using a cell function assay.”
5) Specify the criteria for assessing successful knockdown or overexpression of DYRK3 and PAICS in cell lines.
A5: Thanks to the reviewer's suggestion, we have included criteria specifying the assessment of successful knockout or overexpression of DYRK3 and PAICS in cell lines. Please refer to the revised version.
“4.3. shRNA-Mediated DYRK3 Knockdown and overexpression
We employed a lentiviral technique to reduce DYRK3 expression in SAS and HSC3 cells. Specifically, an shRNA construct was acquired from Origene (Cat#: TL309236V) for this purpose. To create lentiviral particles carrying DYRK3-targeting constructs or scrambled controls, HEK293T cells were transfected with Lipofectamine 3000 (Thermo Fisher Scientific). The resulting viral supernatant was then utilized to infect SAS or HSC3 cells in the presence of polybrene (4 μg/mL; Sigma-Aldrich). Sub-sequently, the transduced cells were cultured in a medium supplemented with puro-mycin (3 μg/mL, Sigma-Aldrich). We utilized data from the pcDNA3.1 mammalian expression vector (Invitrogen, V79020) to formulate polymerase chain reaction (PCR) primers. Supplementary Figure S1 provides the vector map and primer sequences. Ten micrograms of either empty plasmid (pcDNA3.1 vector control plasmid DNA) or the DYRK3 expression plasmid (pcDNA3.1-CMV- DYRK3) were employed. The DNA-lipofectamine reagent complexes were allowed to incubate at room temperature for 30 minutes. Subsequently, this mixture was introduced into the well, and gentle ag-itation was achieved by rocking the plate back and forth. There was no need to remove the reagent complexes after transfection. The cells were then incubated at 37°C in a CO2 incubator for 48 hours. Real-time Quantitative Polymerase Chain Reaction (RT-PCR), along with quantitative Western blotting and detection, are essential tech-niques that can be employed to verify the impact of gene manipulation, such as knockdown or overexpression. RT-PCR is a highly sensitive and precise method that measures the amount of specific RNA, thus allowing for the quantification of gene ex-pression levels following knockdown or overexpression. This is particularly useful in understanding the functional role of specific genes. On the other hand, quantitative Western blotting provides a robust approach to detect and quantify specific proteins, thereby confirming the effects of gene manipulation at the protein level. This combina-tion of RT-PCR and Western blotting offers a comprehensive approach to validate the efficiency and functional consequences of gene knockdown or overexpression in vari-ous biological and clinical research contexts. Successful knockdown or overexpression of cells was confirmed through either quantitative reverse transcription-polymerase chain reaction (qRT-PCR) or Western blotting.”
6) In the mouse models section, elaborate on the rationale for choosing specific treatment groups and provide more details on the experimental design.
A6: Thanks to the reviewer's suggestion, we have included a mouse model section on the rationale for selecting specific treatment groups and provided more details on experimental design. Please refer to the revised version.
“4.11. Mouse Models Utilizing Patient-Derived Xenografts
To establish xenograft models, we employed mice that were homozygous for the severe combined immune deficient (SCID) mutation. We procured twenty-five 8-week-old female mice of the nonobese diabetic (NOD) and SCID strains from Bio-LASCO Taiwan, located in Taipei, Taiwan. These mice were raised under controlled pathogen-free conditions, adhering to the guidelines outlined by the Animal Care and Use Committee of Taipei Medical University Animal Care and User Committee, with an Animal Use Agreement Ratification Affidavit number of Taipei Medical Universi-ty-LAC-2022-0467. In brief, radio-resistance tumors from patients with oral squamous cell carcinoma (OSCC) were placed in RPMI 1640 on ice at the surgical site. The tu-mors were thinly sliced into 2–3 mm3 pieces and rinsed thrice with RPMI 1640. Sub-sequently, these samples were finely minced into fragments small enough to pass through an 18-gauge needle. They were then mixed in a 1:1 ratio (v/v) with Matrigel (obtained from Collaborative Research, Bedford, MA, USA), resulting in a total volume of 0.2 mL for each injection. This tissue mixture was subcutaneously injected into both flanks of 8-week-old male SCID mice. For the experimental groups, we obtained twen-ty 8-week-old female NOD/SCID mice from BioLASCO Taiwan and maintained them in the same controlled pathogen-free conditions. These mice were divided into four groups: Vechical, cisplatin, DYRK3 inhibitor (GSK-626616) or combination therapy, with each group consisting of five mice. The mice received different treatments: vehi-cle (PBS orally five times per week) or GSK-626616 (5mg/kg, orally five times per week). The compound GSK-626616 is a specific inhibitor targeting the dual-specificity tyrosine-regulated kinase 3 (DYRK3), demonstrating potent inhibitory properties with an IC50 value of 0.7 nM. This particular inhibitor is utilized extensively in ani-mal-based experimental research, particularly focusing on tumor treatment. Addition-ally, GSK-626616 plays a critical role in advancing the understanding of molecular in-teractions and mechanisms in cancer biology. It is used to examine the alterations in the network of proteins that interact with DYRK3 when exposed to the GSK-626616 inhibitor. This research is pivotal in comprehending how the inhibition of DYRK3 af-fects cellular processes and contributes to the development of targeted therapies for cancer treatment. Cisplatin, a platinum-containing anti-cancer agent, primarily acts by obstructing DNA synthesis and thereby inhibiting tumor cell growth. Employed ei-ther as a standalone treatment or in conjunction with other medications, it is effective in managing various forms of cancer.In our study, we established animal experimental groups for both single and combined treatment approaches to evaluate the drug's therapeutic capabilities, focusing particularly on the effects of inhibiting the DYRK3 pathway.
To monitor tumor growth, we measured tumor volume using a standard caliper every other week, applying the formula: V = (L × W2)/2, where 'L' represents the long axis of the tumor, and 'W' signifies the width of the tumor. Following the conclusion of the experiments, humane euthanasia was performed on the animals, and tumor and tissue samples were collected for subsequent analyses.”
Please consider the following: In general, the Results and discussion section delves deeply into the investigation of DYRK3 in oral cancer. The use of both whole blood and tumor tissue analysis provides a comprehensive and detailed perspective.
7) However, there's a need to enhance clarity in explaining co-expressed genes, include more statistical details in the kinome siRNA screening, and improve the presentation of results from DYRK3 overexpression. These improvements are essential for making the findings more accessible and engaging for a broader audience.
A7: Thanks to the reviewer's suggestions, we have improved the clarity of interpreting coexpressed genes, included more statistical detail in the kinome siRNA screen, and improved the presentation of DYRK3 overrepresentation results. Please refer to the revised version.
“2.3. Kinome siRNA (small interfering RNA) library screening for kinases in OSCC cells HN12 and SAS cell line were evaluated if any kinase is involved in the radioresistance.
The human kinome comprises around 538 kinase genes, encompassing 89 tyrosine kinases, 429 serine/threonine kinases, and 20 lipid kinases. These kinases are vital for signaling mechanisms that govern a range of cellular physiological activities, in-clud-ing cell proliferation, migration, and differentiation. Notably, tyrosine kinases, among others, are known to play a significant role in the development of tumors, ma-lignan-cies, and resistance to cancer treatments. Consequently, these kinases have be-come fo-cal points for targeted cancer therapies in clinical practices. In our study, we per-formed a comprehensive kinome siRNA (small interfering RNA) library screening to identify kinases that may play a role in radioresistance in OSCC cells. Specifically, we focused on evaluating the HN12 and SAS cell lines to determine the presence of these kinases. To begin the screening process, we seeded the HN12 and SAS (Figure 3A) cells and transfected them with the arrayed kinome siRNA library. In or-der to in-vestigate the kinases that influence cell viability in oral squamous cell carci-noma (OSCC), we conducted a screening of human OSCC cells, specifically the HN12 and SAS cell lines, using an siRNA library that targets 709 human kinases and related genes. Human gingival fibroblasts (HGF-1) served as the control group for this study. From this screening, the top 15 gene targets were initially chosen based on their role in reducing cell proliferation in OSCC cells. Subsequent counter-screening with human gingival fibroblasts (HGF-1) highlighted the impact on several genes, including DYRK3, DDR1, WEE1, and CDC2L2. Focusing on these findings, DYRK3 was selected for fur-ther validation to assess both the effectiveness of its knockdown and its influence on cell viability.
Each gene in the library was targeted with pooled siRNAs (5 nM) for 72 hours. To assess cell viability, we conducted an SRB-assay. Following the initial screening, we identified the top-ranked siRNAs that demonstrated a significant reduction in cell via-bility for both HN12 and SAS cells. To validate the efficiency of the siRNA knock-down, we performed qRT-PCR analysis to measure the mRNA levels of the tar-geted genes, using scrambled siRNA as a control. The fold change in mRNA levels for each siRNA knockdown is indicated above the corresponding bar. Among the various genes that exhibited knockdown, we selected DYRK2 and DYRK3 for further investiga-tion (Figure 3B). Notably, DYRK3 was specifically chosen for subsequent studies. To ensure that the selected genes do not impact normal physiological processes, we ex-tended our analysis to include HGF-1 cells, which are normal human gingival cells. Our observa-tions indicated that the knockdown of these selected genes resulted in re-duced viabil-ity specifically in cancer cells, without affecting the viability of normal HGF-1 cells. This suggests that DYRK3 may hold importance and warrant further in-vestigation. Finally, we used the GSE42743 data set of Gene Expression Omnibus (GEO) to analyze the expression profile of DYRK2 and DYRK3 genes in oral cancer and adja-cent normal tissues. Gene expression profiling involves assessing the array of genes that are tran-scribed under certain conditions or within a particular cell type. This pro-cess provides a comprehensive overview of cellular functions by mapping out gene expression pat-terns. (Figure 3C).
Our study sheds light on the pivotal function of the DYRK3 gene, particularly its role as a crucial kinase in the context of oral squamous cell carcinoma (OSCC). This significance is observed across different cell types within OSCC, encompassing both radioresistant (HN12) and radiosensitive (SAS) cells. Intriguingly, we found that de-pleting DYRK3 not only affects the viability of radioresistant cell lines but also has a marked impact on radiosensitive cell lines. DYRK3 plays a multifaceted role in the survival and proliferation of OSCC cells, regardless of their radiation response char-acteristics. The fact that DYRK3 depletion impacts both types of cell lines hints at its overarching influence in the cellular processes of OSCC. By understanding the spe-cific functions and impacts of genes like DYRK3, we can better tailor treatments to combat OSCC more efficiently, potentially improving patient outcomes in both radio-resistant and radiosensitive scenarios.”
Conclusions: Overall, this paper's strategy was quite successful. To engage readers effectively, consider expanding on future research directions, discussing potential clinical implications, and highlighting the broader impact of your findings on OSCC understanding and treatment. This will enhance the overall appeal and significance of your study.
Thanks to the reviewer's suggestion, we have emphasized discussing potential clinical implications and highlighting the broader implications of the kinase for the understanding and treatment of radioresistance in OSCC. Please refer to the revised version.
“3. Discussion
Radiotherapy is a widely used treatment option for advanced-stage cancer, with about half of all cancer patients receiving this alongside primary treatments like sur-gery and chemotherapy[19]. This method employs high-dose ionizing radiation to in-duce cell death by damaging cellular DNA. However, resistance to radiation, also known as radiation resistance, poses a significant clinical challenge for various cancer types including glioblastoma, breast cancer, and esophageal adenocarcinoma[20]. Certain organelles, especially mitochondria, play pivotal roles in influencing a tumor's response to radiation as they regulate several processes tied to radioresistance[21]. Within the framework of purine metabolism, a key component known as the purino-some has been identified. The formation of the purinosome closely correlates with the cell cycle, presenting a new therapeutic opportunity for treating cancers by targeting purinosome formation and purine metabolism[22]. Recent research findings further underscore the importance of purine synthesis in the aggressive behavior exhibited by glioblastoma (GBM) and other cancer types[23]. Elevated rates of de novo purine and pyrimidine synthesis aid in maintaining glioma-initiating cells, potentially contrib-uting to therapy resistance and recurrence in GBM. While cancer metabolism has at-tracted increasing attention in recent years, the link between metabolism and DNA damage/DNA repair in cancer still requires further investigation [24]. Thus, this report aims to shed light on the association between DNA repair, DNA damage, and purine metabolism.
PAICS, an enzyme involved in the metabolic pathway of purine nucleotides, is a key contributor to the proliferation and spread of tumors [25]. This dual-function en-zyme, which hosts phosphoribosylaminoimidazole carboxylase activity in its N-terminal region and phosphoribosylaminoimidazole succinamide synthetase activi-ty in its C-terminal region, mediates the sixth and seventh stages of purine biosynthe-sis [26]. Notably, PAICS is pivotal in pancreatic ductal adenocarcinoma (PDAC) and is thus regarded as a promising therapeutic target due to its susceptibility to small mole-cule targeting [27]. Despite its recent emergence as a potential target, its role in radia-tion-resistant oral cancer has not been extensively discussed. DYRK, standing for dual specificity tyrosine-regulated kinase, is a protein kinase family conserved across mul-tiple species. It has the capability to phosphorylate numerous proteins integral to a va-riety of cellular processes, including proliferation of cancer cells, enhanced drug re-sistance, and signaling pathways linked to tumor progression[13]. Abnormal regula-tion or expression of DYRK kinase is associated with diverse human diseases, including cancer. Dual-specificity tyrosine phosphorylation-regulated kinase 1A (DYRK1A), a protein kinase that is preserved across evolutionary scales, plays a crucial role in the onset of Down syndrome and other diseases. It influences neurodevelopment, cell pro-liferation and differentiation, tumor formation, and the pathogenesis of neurodegen-erative diseases. Furthermore, DYRK1A is critical in the pathogenesis of various dis-eases and the regulation of signal pathways. Elevated levels of DYRK1B protein and mRNA in tissues of colon adenocarcinoma are essential for tumor initiation and pro-gression, which emphasizes the importance of DYRK1B detection for prognosis pur-poses. Studies on cervical cancer revealed that the expression of DYRK1B protein in-creases in parallel with the progression of cervical lesions, showing a high expression in cervical cancer tissues and cells. Increased levels of DYRK2 in patients are correlat-ed with a better prognosis, improved responses to chemotherapy, and increased sur-vival rates, particularly in instances of liver metastasis originating from colorectal cancer. In contrast, in ovarian serous adenocarcinoma, diminished DYRK2 expression is associated with a poorer prognosis [16].
In cases of hepatocellular carcinoma (HCC), DYRK3 expression was significantly lower compared to normal controls[28]. However, the introduction of DYRK3 in HCC cells led to a marked reduction in tumor growth and metastasis in xenograft tumor models. As such, deciphering the molecular mechanisms that control DYRK can offer fresh perspectives on tumor cell plasticity and responses to both traditional and new treatments. This understanding could pave the way for the identification of novel drug targets for a more efficient, less toxic, and personalized therapy approach for patients with radiation-resistant oral squamous cell carcinoma (OSCC).
The purine biosynthetic pathway has recently emerged as an essential provider of metabolic intermediates for cancer-related processes. There's a growing body of evi-dence pointing to the multifaceted roles of PICAS/DYRK in numerous cellular and physiological responses across various cell types. Despite this, our understanding of the regulation of purine metabolism in Oral Squamous Cell Carcinoma (OSCC) re-mains quite limited. DYRK1A, 1B, 2, and 3 belong to the Dual-specificity tyrosine phosphorylation-regulated kinase (DYRK) family. Our research indicates that DYRK3 expression is notably higher in human cancer tissues than in healthy ones.
Our hypothesis posits that the cancer-inducing role of PAICS/DYRK3 could be in-strumental in fostering radioresistance and metastasis in OSCC by boosting the func-tionality of the purinosome, which in turn enhances the stemness, tumorigenicity, and drug resistance of OSCC. However, a decrease in PAICS expression might potentially counteract these effects. We propose a new PAICS/DYRK3 signaling pathway that could activate purinosome formation reprogramming, which may serve as a targeted treatment for radioresistant OSCC. Purine metabolism is a critical process for tumor development, with PAICS able to trigger cancer purine metabolism and thus facilitate tumor progression. The results obtained from our experimental investigations have provided compelling evidence supporting the critical involvement of DYRK3 and PAICS in oral squamous cell carcinoma (OSCC). Notably, both OSCC patients and cell lines demonstrated a pronounced overexpression of DYRK3, which exhibited a strong correlation with PAICS expression (Table 1, Figure 1 and 2). These findings further substantiate the significance of DYRK3 and PAICS in OSCC biology. Importantly, the upregulation of DYRK3 and its correlation with PAICS expression were found to be associated with a poor prognosis in OSCC patients (Figure 1). This observation under-scores the potential clinical relevance of DYRK3 and PAICS as prognostic markers in OSCC. Moreover, our findings shed light on the proposed involvement of the PAICS/DYRK3 axis in purinosome formation, specifically in the context of radiore-sistance and metastasis development in OSCC (Figure 5). This novel insight suggests that the dysregulation of the PAICS/DYRK3 axis may contribute to the acquisition of radioresistance and the metastatic potential of oral squamous cell carcinoma. These results emphasize the potential significance of the PAICS/DYRK3 axis in orchestrating the modulation of treatment effectors, thus affecting therapeutic outcomes in OSCC. Additionally, the dysregulation of this axis appears to play a complex role in tumor-igenicity, disease progression, and the evasion of therapy in patients with oral squa-mous cell carcinoma. Lastly, we suggest that reducing PAICS expression could lessen the invasion and migration capabilities of radio-resistant OSCC, and combining this with cisplatin treatment could enhance apoptosis in radio-resistant OSCC (Figure 3 and 4). Taken together, our findings highlight the crucial involvement of DYRK3 and PAICS in OSCC pathogenesis. This knowledge opens new avenues for further research on the PAICS/DYRK3 axis as a potential therapeutic target and may ultimately con-tribute to the development of novel treatment strategies for patients with oral squa-mous cell carcinoma. Indeed, when cancer cells were exposed to a dual treatment comprising methotrexate, a well-established inhibitor of purine biosynthesis, and an Hsp90 inhibitor, it resulted in a synergistic cytotoxic impact [29]. While numerous ki-nases have been identified as critical players in cancer malignancy management, the specific role of kinases in promoting radioresistance in oral squamous cell carcinoma (OSCC) remains underexplored. In our study, we conducted a targeted kinome analy-sis for OSCC and pinpointed DYRK3 as a key factor in fostering radioresistance. We theorize that radiation-triggered upregulation of DYRK3 impacts the dynamics of purinosome formation, leading to metabolic shifts and heightened invasiveness in OSCC post-radiation treatment. Considering these roles of DYRK3, we suggest that it could serve as a potential therapeutic target to inhibit the progression of OSCC following ra-diotherapy. This research confirms the potential of the purinosome as a promising target for innovative cancer chemotherapy and introduces the prospect of creating novel combination therapies that hinder purine biosynthesis by targeting both specific enzymes within the pathway and the formation of the purinosome complex.”
8) The conclusion could benefit from expanding on future directions, potential impact, and clinical implications. Suggest discussing avenues for future research, emphasizing the broader significance of their findings for advancing OSCC understanding and potential therapeutic strategies. Encourage a more detailed exploration of implications to enhance the conclusion's depth and completeness.
A8: Thanks to the reviewers' suggestions, we have highlighted the broader implications of these findings for advancing the understanding and potential treatment strategies of OSCC. More detailed exploration of meaning is encouraged to enhance the depth and completeness of conclusions. Please refer to the revised version.
“5. Conclusion
In conclusion, as shown in pictorial abstract of Figure 7. Our experimental results highlight the pivotal roles of DYRK3 and PAICS in oral squamous cell carcinoma. The observed overexpression of DYRK3 and its correlation with PAICS, along with their association with a poor prognosis, underscore their importance in OSCC. Additionally, the proposed involvement of the PAICS/DYRK3 axis in purinosome formation sheds light on its role in the development of radioresistance and metastasis. This dysregula-tion of the PAICS/DYRK3 axis serves as a key modulator, influencing treatment effec-tors and contributing to tumorigenicity, disease progression, and therapy evasion in patients with oral squamous cell carcinoma. In this research, we explore new ki-nase-driven metabolic changes that contribute to the development of radioresistance and increased malignancy in oral squamous cell carcinoma (OSCC). Our findings indicate that radiation exposure triggers an increase in DYRK3 expression in OSCC cells. Furthermore, we observed that suppressing DYRK3 reduces the expression of several genes including PPAT, PAICS, GIT2, ATM, and EGFR, which are linked to purinosome formation and metabolism. Notably, actively reducing DYRK3 expression following radiation exposure markedly decreases the migration and invasion capabilities of OSCC cells. Based on these findings, we propose that targeting DYRK3 could emerge as a novel approach to counteract OSCC malignancy by influencing purinosome for-mation and metabolism.”
